# Enhanced functional detection of synaptic calcium-permeable AMPA receptors using intracellular NASPM

Ian Coombs, Cécile Bats, Craig A Sexton, Dorota Studniarczyk, Stuart G Cull-Candy*, Mark Farrant*

Department of Neuroscience, Physiology and Pharmacology, University College London, London, United Kingdom

**Abstract** Calcium-permeable AMPA-type glutamate receptors (CP-AMPARs) contribute to many forms of synaptic plasticity and pathology. They can be distinguished from GluA2-containing calcium-impermeable AMPARs by the inward rectification of their currents, which reflects voltage-dependent channel block by intracellular spermine. However, the efficacy of this weakly permeant blocker is differentially altered by the presence of AMPAR auxiliary subunits – including transmembrane AMPAR regulatory proteins, cornichons, and GSG1L – which are widely expressed in neurons and glia. This complicates the interpretation of rectification as a measure of CP-AMPAR expression. Here, we show that the inclusion of the spider toxin analog 1-naphthylacetyl spermine (NASPM) in the intracellular solution results in a complete block of GluA1-mediated outward currents irrespective of the type of associated auxiliary subunit. In neurons from GluA2-knockout mice expressing only CP-AMPARs, intracellular NASPM, unlike spermine, completely blocks outward synaptic currents. Thus, our results identify a functional measure of CP-AMPARs, that is unaffected by their auxiliary subunit content.

*For correspondence:
s.cull-candy@ucl.ac.uk (SGC-C);
m.farrant@ucl.ac.uk (MF)

**Competing interest:** The authors declare that no competing interests exist.

## Editor's evaluation

AMPA-type glutamate receptors that lack the GluA2 subunit are calcium-permeable and contribute to calcium influx in plasticity and disease. The authors of this Tools and Resources manuscript describe a method for evaluating the presence of GluA2-lacking receptors using intracellular NASPM that avoids complications related to auxiliary subunits that affect biophysical properties. The compelling results provide a valuable new approach for unambiguously differentiating the presence of Ca-permeable and -impermeable AMPA receptors.

## Introduction

AMPA-type glutamate receptors (AMPARs) mediate the fast component of excitatory postsynaptic currents (EPSCs) throughout the mammalian brain (*Baranovic and Plested, 2016*; *Greger et al., 2017*; *Hansen et al., 2021*) and their regulation allows lasting changes in synaptic strength that are essential for normal brain function (*Huganir and Nicoll, 2013*). AMPARs are cation-permeable channels that exist as homo- or heterotetrameric assemblies of the homologous pore-forming subunits GluA1-4, encoded by the genes *Gria1-4*. While a majority of central AMPARs are GluA2-containing di- or tri-heteromeric assemblies (*Lu et al., 2009*; *Wenthold et al., 1996*; *Zhao et al., 2019*), those lacking GluA2 constitute an important functionally distinct and widely occurring subtype. Editing of *Gria2* pre-mRNA at codon 607 results in the substitution of a genetically encoded glutamine (Q) with an arginine (R). This switch from a neutral to a positively charged residue in the pore-forming loop

causes receptors containing Q/R edited GluA2 to have a greatly reduced $Ca^{2+}$ permeability compared to those lacking GluA2 (*Burnashev et al., 1992*; *Kuner et al., 2001*; *Sommer et al., 1991*). As this mRNA editing is essentially complete, AMPARs are commonly divided into GluA2-containing $Ca^{2+}$-impermeable (CI-) and GluA2-lacking $Ca^{2+}$-permeable (CP-) forms (*Bowie, 2012*; *Burnashev et al., 1992*; *Cull-Candy et al., 2006*).

Aside from allowing $Ca^{2+}$ flux, CP-AMPARs differ from CI-AMPARs in having greater single-channel conductance (*Benke and Traynelis, 2018*, *Swanson et al., 1997*) and susceptibility to voltage-dependent block by the endogenous intracellular polyamines spermine and spermidine (*Bowie and Mayer, 1995*; *Donevan and Rogawski, 1995*; *Kamboj et al., 1995*; *Koh et al., 1995a*). They can also be blocked in a use-dependent manner by extracellular application of the same polyamines (*Washburn and Dingledine, 1996*) or various exogenous organic cations, such as the spermine analog *N*-(4-hydroxyphenylpropanoyl)-spermine (HPP-SP; *Washburn and Dingledine, 1996*), the polyamine-amide wasp toxin philanthotoxin-4,3,3 (PhTx-433; *Washburn and Dingledine, 1996*), the spider toxin analog 1-naphthylacetyl spermine (NASPM; *Tsubokawa et al., 1995*), and the dicationic adamantane derivative IEM-1460 (*Magazanik et al., 1997*).

Block of CP-AMPARs by intracellular polyamines results in inwardly- or bi-rectifying current-voltage relationships. This characteristic rectification – seen during whole-cell patch-clamp recordings in the presence of residual endogenous polyamines or added exogenous spermine – has been utilized extensively to identify the presence of CP-AMPARs in neurons. Importantly, this approach has enabled the identification of cell- and synapse-specific CP-AMPAR expression (*Koh et al., 1995b*, *Tóth and McBain, 1998*), changes in CP-AMPAR prevalence during development (*Brill and Huguenard, 2008*; *Kumar et al., 2002*; *Soto et al., 2007*; *Yang et al., 2010*), and roles for CP-AMPARs in multiple synaptic plasticities, including long-term potentiation (LTP) and depression (LTD) (*Fortin et al., 2010*; *Lamsa et al., 2000*; *Liu and Cull-Candy, 2000*; *Mahanty and Sah, 1998*; *Manz et al., 2020*; *Plant et al., 2006*; *Sanderson et al., 2016*). In addition, its use has suggested CP-AMPAR changes associated with various conditions, including ischemia (*Dixon et al., 2009*; *Liu et al., 2004*; *Peng et al., 2006*), brain trauma (*Korgaonkar et al., 2020*), schizophrenia (*Druart et al., 2021*), stress (*Kuniishi et al., 2020*), and glaucoma (*Sladek and Nawy, 2020*), and in circuit remodeling associated with fear-related behaviors (*Clem and Huganir, 2010*; *Liu et al., 2010*), drug addiction (*Bellone and Lüscher, 2006*; *Conrad et al., 2008*; *Lee et al., 2013*; *Parrilla-Carrero et al., 2021*; *Scheyer et al., 2014*; *Van den Oever et al., 2008*), and neuropathic or inflammatory pain (*Goffer et al., 2013*; *Katano et al., 2008*; *Park et al., 2009*; *Sullivan et al., 2017*; *Vikman et al., 2008*).

Native AMPARs co-assemble with various transmembrane auxiliary subunits that influence receptor biogenesis, synaptic targeting, and function (*Greger et al., 2017*; *Jackson and Nicoll, 2011b*, *Schwenk et al., 2019*). Importantly, several of these, including transmembrane AMPAR regulatory proteins (TARPs), cornichons, and germ cell-specific gene 1-like protein (GSG1L), have been shown to modify the block of CP-AMPARs by intracellular spermine. In the case of TARPs and CNIHs inward rectification is reduced (*Brown et al., 2018*; *Cho et al., 2007*; *Coombs et al., 2012*; *Soto et al., 2007*; *Soto et al., 2009*), whereas with GSG1L the rectification is increased (*McGee et al., 2015*). This can complicate the interpretation of measures of spermine-dependent rectification. Moreover, any change in a rectification that might be attributed to changes in the prevalence of CP-AMPARs could instead reflect a change in auxiliary subunit content.

Here, we show that intracellular NASPM, PhTx-433, and PhTx-74 can be used to specifically and voltage-dependently block recombinant CP-AMPARs. Unlike spermine, which is permeant, these blockers allow negligible outward current at positive potentials. At +60 mV, 10 μM intracellular NASPM produces a use-dependent block, while at 100 μM it fully blocks outward currents in a use-independent manner. Critically, this block is unaffected by the presence of auxiliary subunits. Furthermore, in three neuronal populations from GluA2-knockout (GluA2 KO) mice (*Jia et al., 1996*), known to express different auxiliary proteins (*Fukaya et al., 2006*; *Khodosevich et al., 2014*; *Tomita et al., 2003*; *Yamazaki et al., 2010*), we show that 100 μM intracellular NASPM causes full rectification of glutamate-evoked currents from extrasynaptic and synaptic CP-AMPARs, something that cannot be achieved with spermine. Together, our results reveal that the use of intracellular NASPM provides a simple and effective method for determining the relative contribution of CP-AMPARs to AMPAR-mediated currents, regardless of their associated auxiliary subunits.

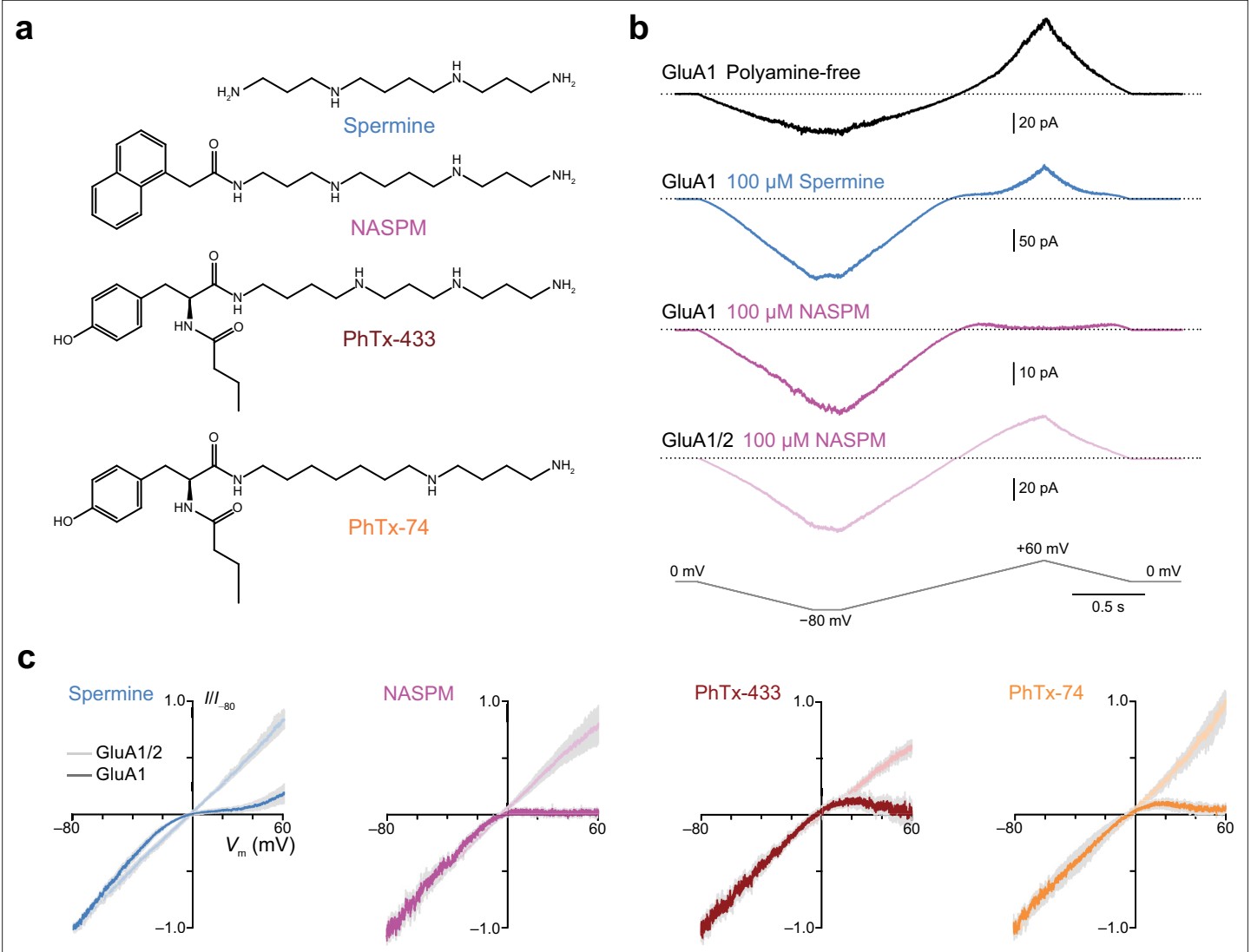

**Figure 1.** Intracellular 1-naphthylacetyl spermine (NASPM) and polyamine toxins specifically block GluA2-lacking calcium-permeable AMPA-type glutamate receptors (CP-AMPARs) in a voltage-dependent manner. (**a**) The blockers used in this study. (**b**) Representative responses activated by 300 μM glutamate and 50 μM cyclothiazide from outside-out patches excised from HEK293 cells expressing either GluA1 or GluA1/2. The voltage was ramped linearly from −80 to +60 mV (100 mV/s). GluA1 displayed outward rectification with a polyamine-free pipette solution. In the presence of 100 μM spermine GluA1 displayed a doubly rectifying relationship, and with 100 μM NASPM GluA1 displayed full inward rectification. GluA1/2 did not rectify in the presence of NASPM. (**c**) Normalized and pooled current-voltage (*I-V*) relationships for GluA1 and GluA1/2 in the presence of 100 μM intracellular polyamines (n=3–8). Colored traces denote the mean and gray shading ± standard error of the mean. For all blockers the RI$_{+60/−60}$ value with GluA1 was less than that with GluA1/2; the unpaired mean differences and 95% confidence intervals were −0.85 [−1.09,−0.63] with spermine (n=4 GluA1/2 patches and 5 GluA1 patches), −1.07 [−1.7,−0.79] with NASPM (n=8 and 4),−0.82 [−0.96,−0.60] with PhTx-433 (n=3 and 4), and −0.78 [−0.91,−0.70] with PhTx-74 (n=3 and 4).

The online version of this article includes the following source data for figure 1:

**Source data 1.** RI$_{+60/−60}$ values from GluA1 and GluA1/γ2 ramp current-voltage (*I-V*) relationships (300 μM glutamate) were recorded with intracellular spermine, NASPM, PhTx-433, or PhTx-74 (each 100 μM).

## Results

### Intracellular polyamine toxins block CP-AMPARs

To determine whether intracellularly applied NASPM or polyamine toxins might offer advantages over spermine for the identification of native CP-AMPARs we examined three compounds: NASPM, which contains the same polyamine tail as spermine, PhTx-433, which has a different distribution of amines, and PhTx-74, which lacks one amine group (*Figure 1a*). Initially, we recorded currents in outside-out

patches from HEK cells transiently transfected with GluA1 alone or with GluA1 and GluA2, to produce homomeric CP- and heteromeric CI-AMPARs, respectively. The receptors were activated by glutamate (300 µM) in the presence of cyclothiazide (50 µM) to minimize AMPAR desensitization and we applied voltage ramps (100 mV/s) to generate current-voltage (*I-V*) relationships.

As expected (*McGee et al., 2015*; *Soto et al., 2007*), in the absence of intracellular polyamines homomeric GluA1 receptors generated outward currents at positive potentials, showing clear outward rectification, while in the presence of 100 µM spermine, they displayed doubly rectifying responses (*Figure 1b*). By contrast, when the intracellular solution contained 100 µM NASPM, GluA1 receptors displayed inwardly rectifying responses with negligible current passed at positive potentials (*Figure 1b*). Unlike responses from GluA1 alone, currents from GluA1/2 receptors in the presence of NASPM were non-rectifying (*Figure 1b*). *I-V* plots showed that, when added to the intracellular solution at 100 µM, each blocker conferred marked inward rectification on the currents mediated by GluA1 (rectification index, $RI_{+60/-60}$ 0.02–0.26), but not on those mediated by GluA1/2 ($RI_{+60/-60}$ 0.84–1.30) (*Figure 1c*). Thus, although differing in structure, they all produced selective voltage-dependent block of the GluA2-lacking CP-AMPARs. Of note, the block by intracellular PhTx-74 was restricted to CP-AMPARs, despite the fact that it produces low-affinity block of CI-AMPARs when applied extracellularly (*Jackson et al., 2011b*; *Nilsen and England, 2007*).

We next investigated the concentration- and auxiliary subunit-dependence of the block. Specifically, we generated conductance-voltage (*G-V*) relationships and fit those from inwardly rectifying responses with a single Boltzmann function and those from doubly rectifying responses with a double Boltzmann function (*Panchenko et al., 1999*). This revealed that as the concentration of added blocker was increased (from 0.1 or 1 µM to 500 µM) there was a progressive negative shift in $V_b$ (the potential at which 50% block occurs) (*Figure 2a and b*). Plotting $V_b$ against polyamine concentration (*Figure 2b*) allowed us to determine the $IC_{50, 0\,mV}$ (the concentration expected to result in a half maximal block at 0 mV) and thus estimate the potency of each polyamine. This showed that, for steady-state conditions, the order of potency for the GluA1 block was spermine >NASPM >PhTx-433 >PhTx-74 (*Figure 2b and c*). The same analysis demonstrated that the potency of each blocker was reduced when GluA1 was co-expressed with TARP γ2 (between 7- and 18-fold reduction; *Figure 2b and c*).

## Onset and recovery of the block by NASPM

Next, we examined GluA1/γ2 currents elicited by rapid application of 10 mM glutamate (in the absence of cyclothiazide) and compared the effect of NASPM – the second most potent of the blockers – with that of spermine. We first tested the blockers at a concentration of 10 µM, as this allowed us to examine the onset of the block. At both positive and negative voltages, glutamate application produced currents that showed a clear peak and rapid desensitization to a steady-state level. However, with intracellular NASPM the steady state-currents at positive voltages were much reduced compared to the corresponding currents at negative voltages (*Figure 3a*). For both peak and steady-state currents the $RI_{+60/-60}$ values obtained with NASPM were less than those obtained with spermine. For peak current, $RI_{+60/-60}$ was 0.37 ± 0.04 for NASPM and 0.56 ± 0.07 for spermine (n=11 and 10, respectively; unpaired mean difference −0.19 [−0.34,−0.045], p=0.031, two-sided Welch two-sample *t*-test). For steady-state current, $RI_{+60/-60}$ was 0.04 ± 0.02 for NASPM and 0.90 ± 0.14 for spermine (n=8 and 7), respectively (unpaired mean difference −0.86 [−1.12,−0.62], p=0.00075). This difference was also evident in the *G-V* relationships; those determined from peak responses in the presence of spermine or NASPM were largely overlapping, while those from steady-state responses were markedly different at potentials positive to +40 mV, with a large outward conductance seen in the presence of spermine but not NASPM (*Figure 3b*).

Intracellular spermine is a weakly permeable open-channel blocker of both CP-AMPARs and kainate receptors (KARs) (*Bowie et al., 1998*; *Brown et al., 2016*). However, for GluK2(Q) KARs, permeation of intracellular polyamines has been shown to vary with molecular size, with two molecules of greater width than spermine – *N*-(4-hydroxyphenylpropanoyl)-spermine (HPP-SP) and PhTx-433 – showing little or no detectable relief from the block with depolarization (*Bähring et al., 1997*). We reasoned that NASPM, with its naphthyl headgroup, might also be expected to display limited permeability of CP-AMPAR channels. This could account for the shape of the ramp *I-V* (*Figure 1c*) and *G-V* plots with NASPM (*Figure 2a*). Indeed, the pronounced effect of 10 µM intracellular NASPM on steady-state

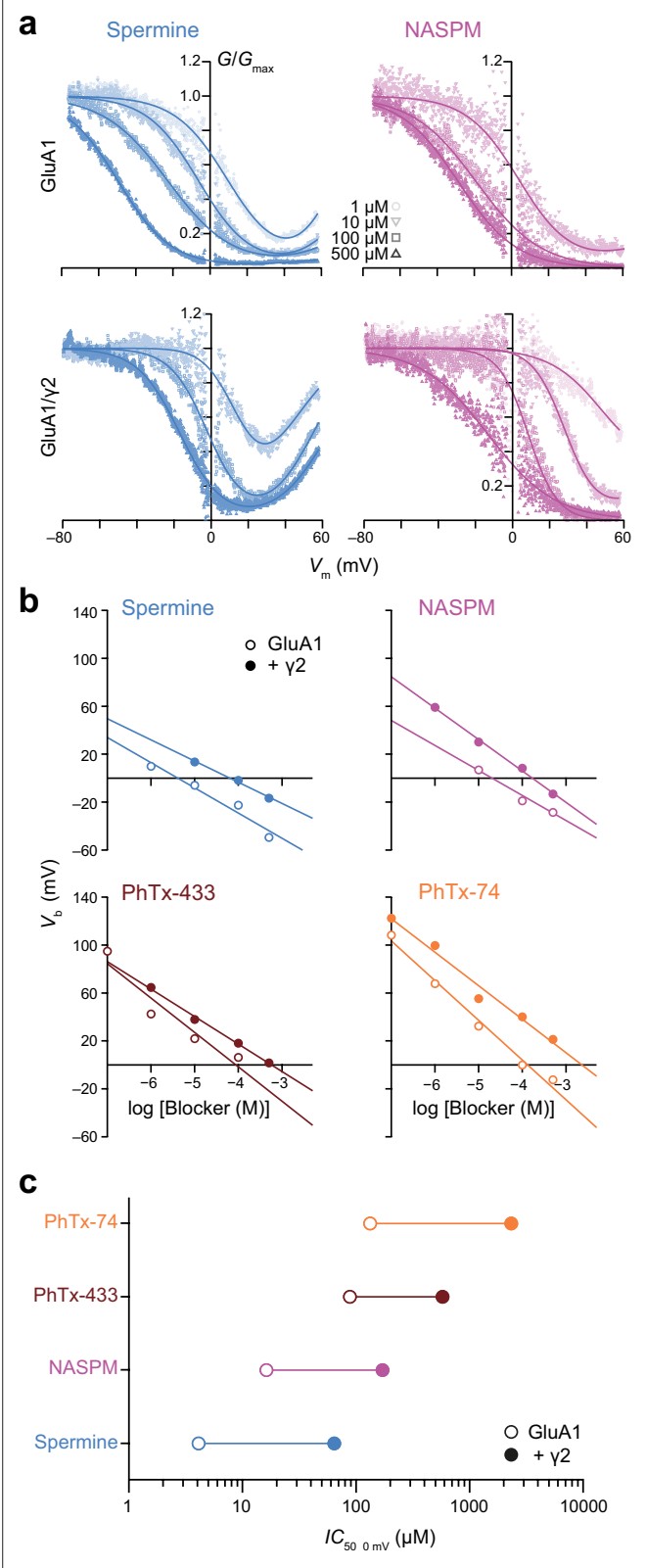

**Figure 2.** Spermine, 1-naphthylacetyl spermine (NASPM), and polyamine toxins display different potencies of GluA1 block that are all decreased by transmembrane AMPAR regulatory proteins (TARP) γ2. (**a**) Pooled, normalized conductance-voltage (*G-V*) relationships (from voltage ramps as in *Figure 1*) for GluA1 and GluA1/γ2 in the presence of spermine (left) or NASPM (right) (n=4–8). Conductance was corrected for the outwardly rectifying

*Figure 2 continued on next page*

*Figure 2 continued*

response seen in polyamine-free conditions and fitted with single or double Boltzmann relationships (solid lines). (**b**) $V_b$ values from fitted *G-V* relationships of GluA1 and GluA1/γ2 in the presence of varying concentrations of blockers. When $V_b$ was plotted against log [blocker] a linear relationship could be fitted in all cases, the x-intercept of which gave $IC_{50,\,0\,mV}$. (**c**) The $IC_{50,\,0\,mV}$ values for each polyamine demonstrate relative potency in the order spermine >NASPM >PhTx-433 >PhTx-74, with γ2 co-expression reducing the potency of the blockers by 7–18-fold.

The online version of this article includes the following source data for figure 2:

**Source data 1.** $V_b$ values for GluA1 and GluA1/γ2 obtained with intracellular spermine, NASPM, PhTx-433, or PhTx-74.

**Source data 2.** $IC_{50\,0\,mV}$ values for GluA1 and GluA1/γ2 obtained with intracellular spermine, NASPM, PhTx-433, or PhTx-74.

GluA1/γ2 currents at positive potentials (*Figure 3a*) is also consistent with limited permeation, leading to the accumulation of channel block.

In line with this, we found that the decay of GluA1/γ2 currents recorded in the presence of NASPM was strongly voltage-dependent (*Figure 3a and c*). At negative potentials, $\tau_{decay}$ values were similar in the presence and absence of NASPM. However, at positive potentials (from +10 to +80 mV) the kinetics in the two conditions differed markedly. In the absence of NASPM current decay was slowed at positive potentials, while in the presence of NASPM, the decay was progressively accelerated (*Figure 3d*). In the absence of NASPM, $\tau_{decay}$ was slower at +60 mV than at −60 mV (7.3 ± 0.6 ms *versus* 5.7 ± 0.3 ms, n=8; paired mean difference 1.55 ms [0.86, 2.62], p=0.019 two-sided paired Welch *t*-test). By contrast, in the presence of NASPM $\tau_{decay}$ was markedly faster at +60 mV than at −60 mV (1.7 ± 0.2 ms *versus* 5.6 ± 0.6 ms, n=10; paired mean difference −4.14 ms [−5.44,−3.07], p=0.00012, two-sided paired Welch *t*-test). Of note, at +60 mV, along with the accelerated decay in the presence of NASPM, we also observed a dramatic slowing of the recovery of peak responses following the removal of glutamate (*Figure 3e*). Currents were elicited by pairs of 100 ms glutamate applications at frequencies from 0.125 to 4 Hz. The peak amplitude of successive responses is normally shaped solely by the kinetics of recovery from desensitization. In the absence of polyamines, a small degree of residual desensitization was apparent at 4 Hz (150 ms interval), but full recovery was seen at all other intervals, at both +60 and −60 mV (*Figure 3f*). However, in the presence of NASPM, although responses at −60 mV were indistinguishable from those in the absence of polyamines, at +60 mV an additional slow component of recovery ($\tau_{rec\,slow}$ 4.9 s; *Figure 3f*) was present. The biphasic recovery is suggestive of two populations of receptors, those that desensitized before being blocked (fast component of recovery), and those that were blocked by NASPM before they desensitized (slow component of recovery). Taken together, our data suggest that 10 μM intracellular NASPM produces a pronounced, rapid and long-lasting inhibition of CP-AMPARs at positive potentials.

## NASPM can induce complete rectification that is unaffected by auxiliary subunits

We found that with 10 μM NASPM, the block of CP-AMPARs was incomplete (*Figure 3a and b*) and depended on the recent history of the channel (*Figure 3f*). However, as intracellular polyamines produce both closed- and open-channel blocks of CP-AMPARs (*Bowie et al., 1998*; *Rozov et al., 1998*), we reasoned that a higher concentration of NASPM would block more effectively and cause pronounced rectification, regardless of recent activity. With a higher concentration of NASPM not only would closed-channel block be more favored, rendering a higher proportion of receptors silent prior to activation, but unblocked closed receptors would more rapidly enter a state of open-channel block once activated at positive potentials. For example, an open channel block with 100 μM would be approximately 10 times faster than with 10 μM NASPM and would be expected to rapidly curtail charge transfer.

Importantly, the block of CP-AMPARs by intracellular spermine is known to be affected by AMPAR auxiliary proteins. Different TARP and CNIH family members reduce inward rectification to varying extents (*Brown et al., 2018*; *Cho et al., 2007*; *Coombs et al., 2012*; *Soto et al., 2007*; *Soto et al., 2009*), while GSG1L increases rectification (*McGee et al., 2015*). Thus, an experimentally observed change in spermine-induced rectification may not arise solely from a change in CP-AMPAR prevalence. Accordingly, we next sought to determine whether intracellular NASPM was able to produce

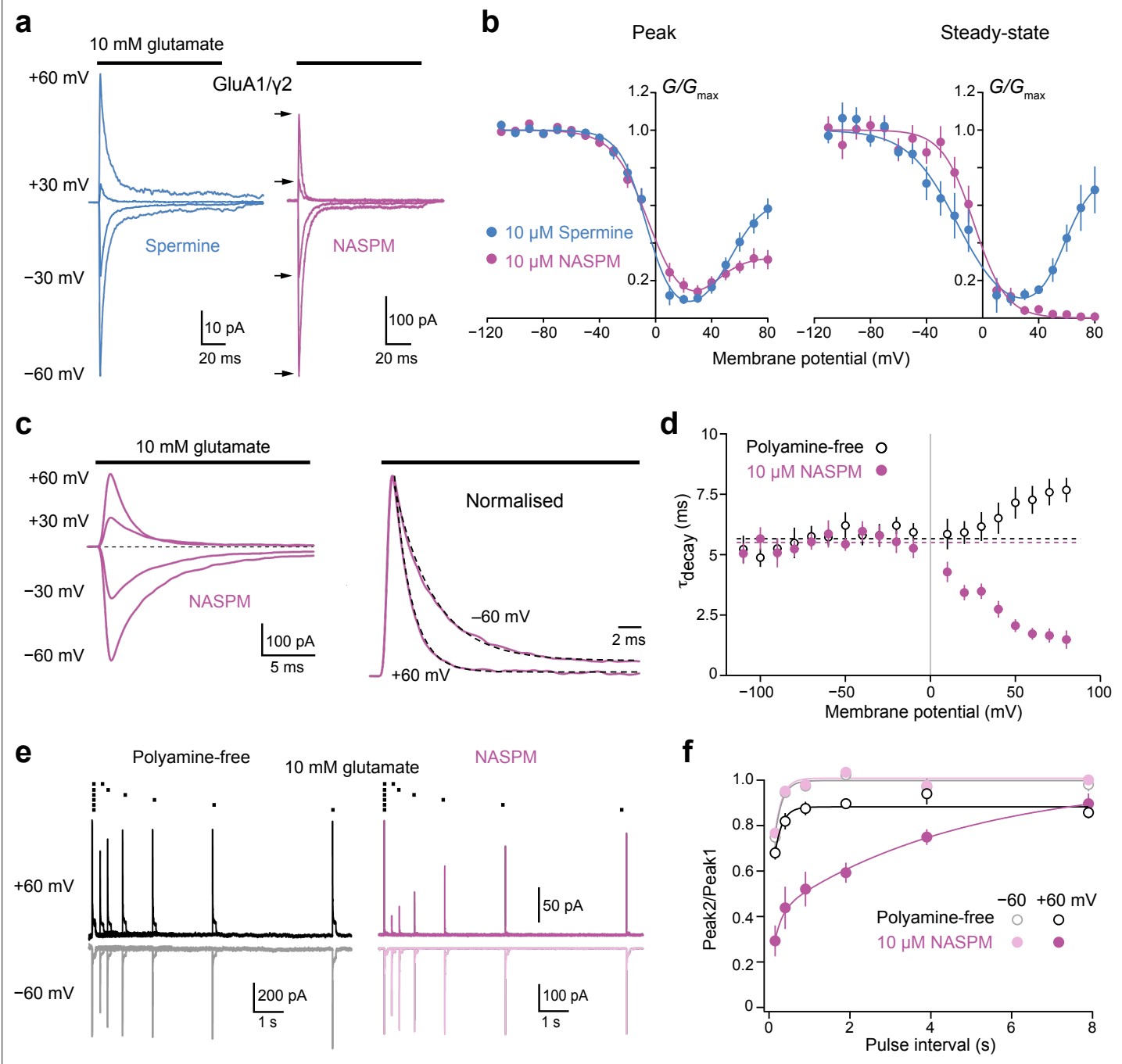

**Figure 3.** Block of GluA1/γ2 by 10 μM intracellular 1-naphthylacetyl spermine (NASPM) shows use-dependence and slow recovery. (**a**) Representative GluA1/γ2 currents in the presence of 10 μM intracellular spermine or NASPM activated by fast applications of glutamate (10 mM, 100 ms) at positive and negative potentials. In the presence of NASPM, relative peak outward currents are smaller than those seen with spermine, while steady-state currents are negligible. (**b**) Pooled, averaged conductance-voltage (*G-V*) relationships for peak and steady-state currents obtained with spermine (n=9 and 6, respectively) and NASPM (n=11 and 8, respectively). Symbols and error bars indicate the mean ± standard error of the mean. Note the stronger inhibition by NASPM of the steady-state compared with peak currents. (**c**) Representative GluA1/γ2 currents with 10 μM intracellular NASPM at the indicated voltages (left) and normalized currents from the same patch at +60 and −60 mV (right). Applications of glutamate at each voltage were preceded by an application at −60 mV to relieve the NASPM block. The dashed lines are single exponential fits. (**d**) Pooled data showing weighted time constants of decay with polyamine-free (open symbols, n=8) and 10 μM NASPM (n=8) intracellular solutions measured between −110 mV and +80 mV. Symbols and error bars indicate the mean ± standard error of the mean. The kinetics of the negative limb (−110 mV to −10 mV) were voltage and NASPM independent (fitted lines indicate $\tau_{decay}$ values of 5.7 ms for polyamine free and 5.4 ms for NASPM). The kinetics of the positive limb (+10 to +80 mV) was markedly accelerated in the presence of intracellular NASPM. (**e**) Superimposed GluA1/γ2 currents in the presence or absence of 10 μM

*Figure 3 continued on next page*

*Figure 3 continued*

NASPM elicited by pairs of glutamate applications at 4, 2, 1, 0.5, 0.25, and 0.125 Hz (10 mM, 100 ms,+/−60 mV). The first responses with NASPM were scaled to those in the polyamine-free condition. With NASPM at −60 mV and in the absence of polyamine at both voltages, the second currents broadly recovered to the initial levels. With NASPM at +60 mV, however, peak currents recovered much more slowly. (**f**) Pooled recovery data recorded in the polyamine-free condition (open symbols, n=9) and with added 10 µM NASPM (filled symbols, n=9). Symbols and error bars indicate the mean ± standard error of the mean.

The online version of this article includes the following source data for figure 3:

**Source data 1.** Normalized *G-V* data for peak and steady-state currents evoked by 10 mM glutamate from GluA1/γ2 receptors with intracellular spermine (10 µM) or NASPM (10 µM).

**Source data 2.** $\tau_{decay}$ values for currents evoked by 10 mM glutamate from GluA1/γ2 receptors with intracellular NASPM (10 µM) or with the polyamine-free intracellular solution at different membrane voltages.

**Source data 3.** Recovery data (Peak 2/Peak 1) for currents evoked by 10 mM glutamate from GluA1/γ2 receptors with intracellular NASPM (10 µM) or with the polyamine-free intracellular solution at +60 and −60 mV.

complete rectification, and whether its action was affected by the presence of different auxiliary subunits.

We first examined the effect of NASPM and spermine on CI-AMPARs. When 100 µM spermine or NASPM was added to the internal solution (*Figure 4a*) the $RI_{+60/−60}$ values were 0.90 ± 0.03 and 0.93 ± 0.06 (both n=6), while with 400 µM the corresponding $RI_{+60/−60}$ values were 0.85 ± 0.11 and 0.77 ± 0.09 (n=5 and 6). This suggests either, that both spermine and NASPM have a small effect on CI-AM-PARs, or that there may have been a minor complement of CP-AMPARs (GluA1 homomers) in some patches. Given that the $RI_{+60/−60}$ values were ≥0.9 with 100 µM spermine or NASPM this concentration was chosen for all subsequent experiments.

We next compared the effects of spermine and NASPM on *I-V* relationships for GluA1 expressed alone or with different auxiliary subunits representing a broad cross-section of those known to differentially modulate the function of native AMPARs (*Figure 4b–d*). Thus, we expressed GuA1 with TARPs (γ2, γ7 or γ8), cornichons (CNIH2 or CNIH3), GSG1L, or CKAMP59. We also examined *I-V* relationships in the presence of γ8 together with CNIH2 or CKAMP44. With GSG1L, full rectification was seen with both 100 µM spermine and 100 µM NASPM ($RI_{+60/−60}$=0). For all other combinations, in the presence of spermine $RI_{+60/−60}$ varied (from 0.037 to 0.29) depending on the auxiliary subunit present, but with NASPM rectification was essentially complete in all cases ($RI_{+60/−60}$ varied from 0 to 0.022) (*Figure 4c*). Thus, unlike 100 µM spermine, 100 µM NASPM produces a near-total block of outward CP-AMPAR-mediated currents that appears to be independent of the type of auxiliary subunit.

## The effects of intracellular NASPM on native AMPARs

To determine whether the difference between spermine and NASPM seen with recombinant receptors was preserved in different native receptor populations, we next compared the actions of the polyamines on extrasynaptic and synaptic receptors in neurons from wild-type and GluA2 KO (*Gria2^tm1Rod*/J) mice (*Jia et al., 1996*).

First, we compared the rectification conferred by 100 µM spermine or NASPM in patches pulled from visually identified dentate gyrus granule cells (DGGC) in acute hippocampal slices (*Figure 5a and b*). These cells strongly express auxiliary subunits CKAMP44 and γ8 (*Fukaya et al., 2006*; *Khodosevich et al., 2014*; *Tomita et al., 2003*) and AMPAR-mediated currents from excised patches have previously been shown to display partial rectification, potentially indicative of a mixed complement of CP- and CI-AMPARs (*Khodosevich et al., 2014*). In wild-type patches, our data were consistent with this. With spermine, the $RI_{+60/−60}$ was 0.62 ± 0.04 (n=6) (range 0.50–0.72) and with NASPM it was 0.58 ± 0.05 (n=8) (range 0.45–0.95). The unpaired mean difference was −0.037 [−0.14, 0.10] (p=0.59, two-sided Welch two-sample *t*-test) (*Figure 5c*). For GluA2 KO DGGC, although both polyamines conferred enhanced rectification, there was a clear difference in its extent. In the presence of spermine, outward currents were present at +60 mV and the $RI_{+60/−60}$ was 0.26 ± 0.08 (n=5) (range 0.15–0.56). By contrast, outward currents were negligible in the presence of NASPM and the $RI_{+60/−60}$ was 0.02 ± 0.01 (n=6) (range 0.003–0.05). The unpaired mean difference was −0.24 [−0.47,−0.15] (p=0.036, two-sided Welch two-sample *t*-test). These data from DGGC patches are consistent with our recombinant data and demonstrate that on native CP-AMPARs NASPM also produces more pronounced rectification than spermine.

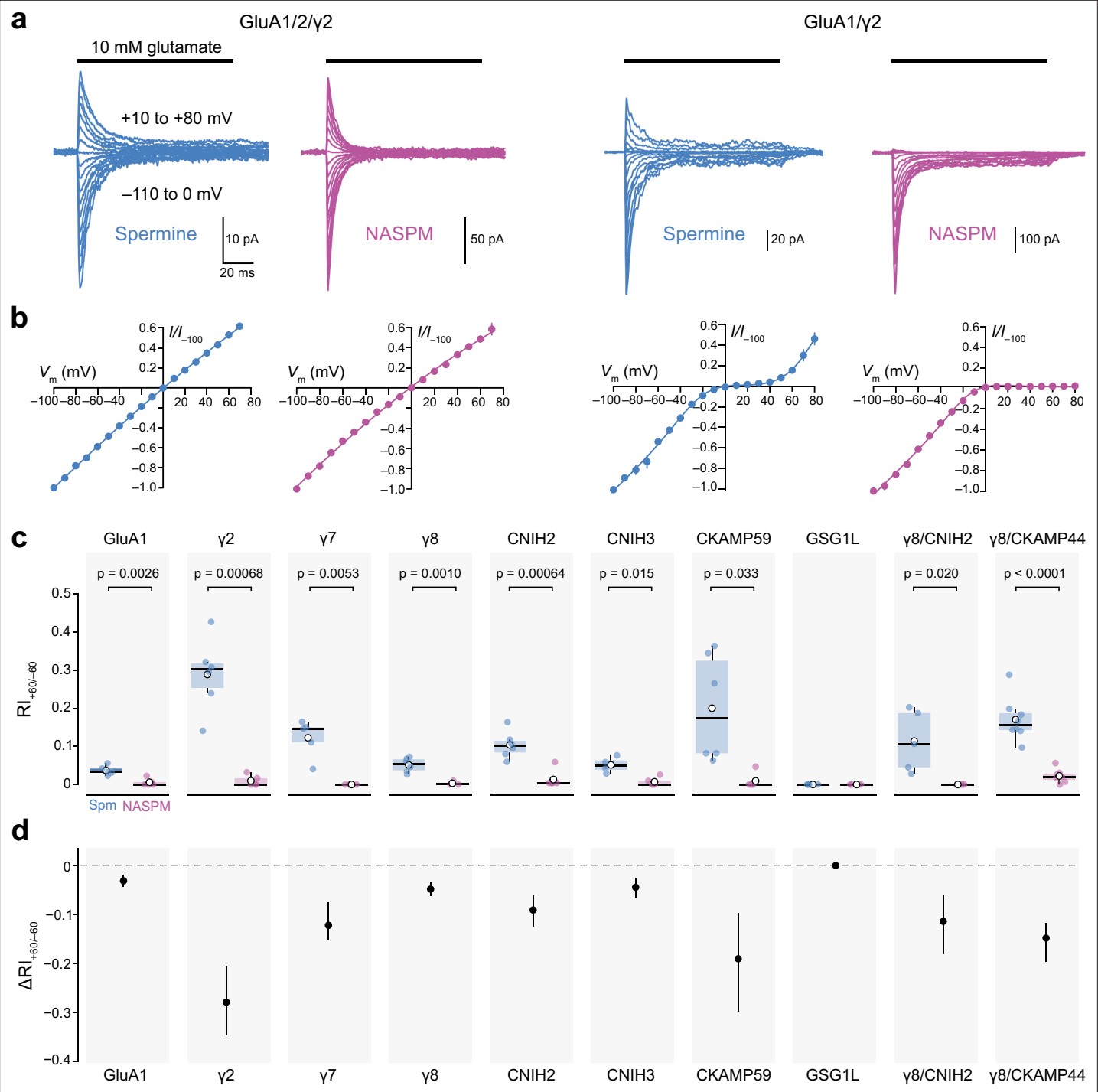

**Figure 4.** Calcium-permeable AMPA-type glutamate receptor (CP-AMPAR) rectification in the presence of 100 μM intracellular 1-naphthylacetyl spermine (NASPM) is unaffected by auxiliary proteins. (**a**) Representative currents activated by fast applications of glutamate to outside-out patches excised from HEK293 cells transfected with GluA1/GluA2/γ2 (left) or GluA1/γ2 (right) (10 mM, 100 ms, −110 to +80 mV). The intracellular solution was supplemented with 100 μM spermine (blue) or NASPM (purple). (**b**) Mean normalized peak current-voltage (*I-V*) relationships for GluA1/GluA2/γ2 and GluA1/γ2 with 100 μM spermine (blue) or NASPM (purple). Symbols and error bars indicate the mean ± standard error of the mean. (**c**) Pooled rectification data (RI$_{+60/−60}$) for GluA1 and GluA1/auxiliary subunit combinations with 100 μM spermine (Spm) or NASPM. Box-and-whisker plots indicate the median (black line), the first and third quartiles (Q1 and Q3; 25–75th percentiles) (box), and the range min(x[x>Q1+1.5 * inter-quartile range]) to max(x[x<Q3+1.5 * inter-quartile range]) (whiskers). Indicated p-values are from two-sided Welch two-sample *t*-tests (note that for GSG1L all RI values were 0). (**d**) Difference plots showing the shift in rectification index (Δ RI$_{+60/−60}$) in the presence of NASPM compared with spermine. Symbols show mean unpaired differences, and the error bars indicate the bootstrapped 95% confidence intervals.

*Figure 4 continued on next page*

*Figure 4 continued*

The online version of this article includes the following source data for figure 4:

**Source data 1.** Normalized current-voltage (*I-V*) data for peak currents evoked by 10 mM glutamate from GluA1/2/γ2, and GluA1/γ2 receptors with intracellular spermine (100 µM) or NASPM (100 µM).

**Source data 2.** $RI_{+60/-60}$ values for peak currents evoked by 10 mM glutamate from GluA1, GluA1/γ2, GluA1/γ7, GluA1/γ8, GluA1/CNIH2, GluA1/CNIH3, GluA1/CKAMP59, GluA1/GSG1L, GluA1/γ8/CNIH2, and GluA1/γ8/CKAMP44 receptors with intracellular spermine (100 µM) or NASPM (100 µM).

Next, we recorded mEPSCs from stellate cells in cerebellar cultures prepared from GluA2 KO mice (*Figure 6a*). Functional studies have shown these molecular layer interneurons normally possess both CI- and CP-AMPARs (*Liu and Cull-Candy, 2000*; *Liu and Cull-Candy, 2002*), formed from GluA2, –3 and –4 subunits (*Yamasaki et al., 2011*) that are expressed along with the TARPs γ2 and γ7 (*Bats et al., 2012*; *Fukaya et al., 2005*; *Yamazaki et al., 2010*). Thus, in stellate cells from GluA2 KO mice, it is expected that AMPAR-mediated EPSCs will be mediated by TARP-associated GluA2-lacking CP-AMPARs.

With 100 µM spermine added to the intracellular solution we detected mEPSCs in stellate cells from GluA2 KO mice at both negative and positive voltages (*Figure 6b and d*). At −60 mV the mean absolute amplitude of the averaged mEPSCs was 39 ± 5 pA, the 20–80% rise time was 0.21 ± 0.02 ms, and the $\tau_{w,decay}$ was 1.14 ± 0.09 ms (n=6). At +60 mV the corresponding measures were 22 ± 2 pA, 0.23 ± 0.03 ms, and 1.58 ± 0.19 ms. In each cell, fewer events were detected at +60 mV than at −60 mV (the mean frequency was 2.3 ± 1.6 Hz at +60 mV and 7.7 ± 5.0 Hz at −60 mV). This is consistent with the anticipated blocking action of spermine rendering a proportion of mEPSCs undetectable at positive potentials (*McGee et al., 2015*).

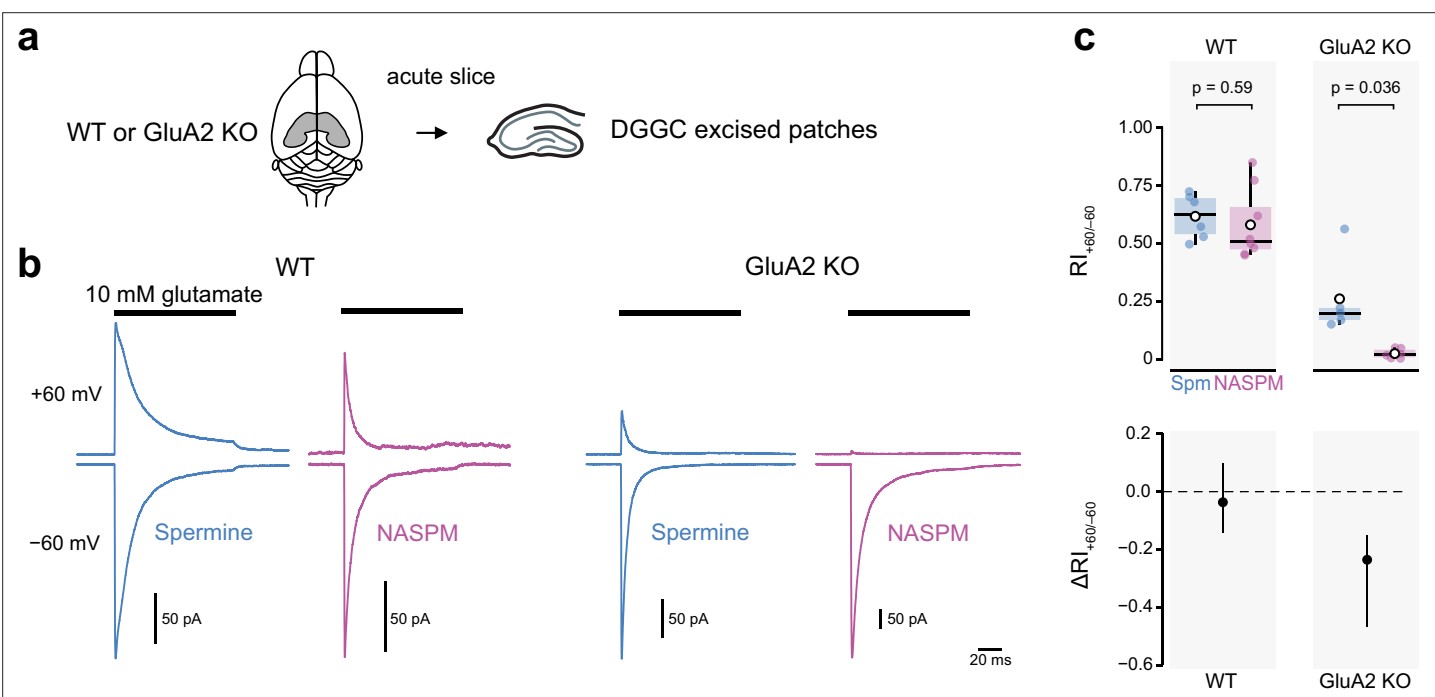

**Figure 5.** Intracellular 1-naphthylacetyl spermine (NASPM) (100 µM) produces full rectification of currents carried by extrasynaptic calcium-permeable AMPA-type glutamate receptors (CP-AMPARs) from dentate gyrus granule cells. (**a**) Diagrammatic representation of the preparation. (**b**) Representative average glutamate-evoked currents from four outside-out patches excised from somata of wild-type (WT) and GluA2 KO dentate gyrus granule cells held at −60 and +60 mV with 100 µM intracellular spermine (blue) or 100 µM intracellular NASPM (purple). (**c**) Pooled rectification data ($RI_{+60/-60}$) for WT and GluA2 KO patches with 100 µM spermine (Spm) or NASPM. Box-and-whisker plots (top) and difference plots showing the shift in rectification index ($\Delta RI_{+60/-60}$) in the presence of NASPM compared with spermine (bottom) as in *Figure 4c and d*. Indicated p-values are from two-sided Welch two-sample *t*-tests.

The online version of this article includes the following source data for figure 5:

**Source data 1.** $RI_{+60/-60}$ values for peak currents evoked by 10 mM glutamate from wild-type (WT) and GluA2 KO DGGCs with intracellular spermine (100 µM) or NASPM (100 µM).

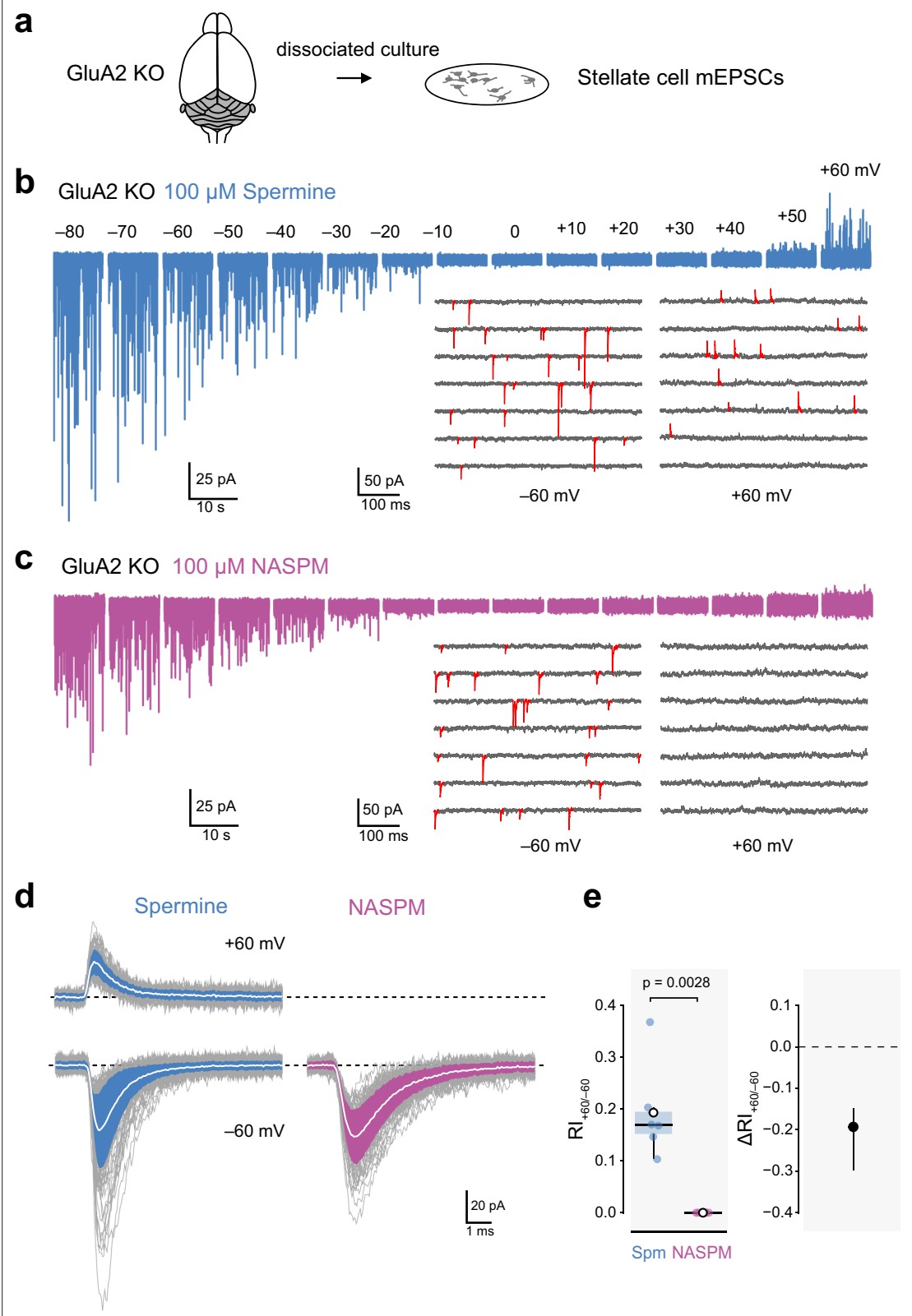

**Figure 6.** Intracellular 1-naphthylacetyl spermine (NASPM) (100 µM) produces full rectification of calcium-permeable AMPA-type glutamate receptor (CP-AMPAR)-mediated mEPSCs. (**a**) Diagrammatic representation of the preparation. (**b**) Representative whole-cell recording from a cultured GluA2-knockout (GluA2 KO) cerebellar stellate cell held at voltages between −80 and +60 mV with 100 µM intracellular spermine. The inset shows expanded sweeps at −60 and +60 mV, with mEPSCs indicated in red. Note the presence of mEPSCs at both voltages. (**c**) Same as a, but for a cell with 100 µM

*Figure 6 continued on next page*

*Figure 6 continued*

intracellular NASPM. The inset shows that mEPSCs (red) were present at −60 mV but not at +60 mV. (**d**) Representative averaged mEPSCs from a cell recorded with intracellular spermine (left) and from a cell recorded with intracellular NASPM (right). Individual mEPSCs are shown in gray, the averaged mEPSCs are shown in white, superimposed on filled areas indicating the standard deviation of each recording. (**e**) Box-and-whisker plot and corresponding difference plot as described in *Figure 4c and d* of pooled $RI_{+60/−60}$ data. Indicated p-value is from a two-sided Welch two-sample *t*-test.

The online version of this article includes the following source data for figure 6:

**Source data 1.** $RI_{+60/−60}$ values for cerebellar stellate cell mEPSCs were recorded with intracellular spermine (100 µM) or NASPM (100 µM).

To take into account the apparent change in frequency when assessing the degree of rectification, we calculated $RI_{+60/−60}$ from the mean absolute amplitude of detected events multiplied by the frequency at each voltage (see Materials and methods). With spermine, the calculated $RI_{+60/−60}$ was 0.19 ± 0.04 (*Figure 6e*). When we made recordings with 100 µM intracellular NASPM, mEPSCs were detectable at negative but not positive voltages (*Figure 6c and d*). At −60 mV the mean absolute amplitude of the averaged mEPSCs was 36 ± 6 pA, the 20–80% rise time was 0.26 ± 0.02 ms, and the $\tau_{w,decay}$ was 1.45 ± 0.17 ms (n=6). At −60 mV the average mEPSC frequency was 14.2 ± 8.2 Hz (n=6), but at +60 mV no events were seen ($RI_{+60/−60}$=0; *Figure 6e*). Thus, even when synapses contained exclusively GluA2-lacking AMPARs, spermine did not fully block outward currents. By contrast, NASPM produced full inward rectification, providing an unambiguous read-out of the presence of a receptor population composed solely of CP-AMPARs.

## NASPM-induced rectification of CP-AMPAR-mediated eEPSCs

To compare the actions of NASPM and spermine on AMPARs at synapses formed in vivo we next examined the effects of the polyamines on the rectification of mossy fiber-evoked excitatory postsynaptic currents (MF-eEPSCs) in granule cells in acute cerebellar slices (*Figure 7a*). Cerebellar granule cells (CGCs) express predominantly GluA2 and GluA4 (*Delvendahl et al., 2019*; *Gallo et al., 1992*; *Mosbacher et al., 1994*), together with TARPs γ2 and γ7 (*Fukaya et al., 2005*; *Tomita et al., 2003*; *Yamazaki et al., 2010*).

We stimulated mossy fibers at 1 Hz and recorded eEPSCs in CGCs from wild-type and GluA2 KO mice at both positive and negative voltages (*Figure 7b and c*). In wild-type CGCs, *I-V* curves were linear (*Figure 7d*), with spermine and NASPM yielding similar $RI_{+60/−60}$ values of 1.07 ± 0.05 (n=5) and 0.94 ± 0.05 (n=7), respectively. The unpaired mean difference was −0.13 [−0.26,–0.009] (p=0.096, two-sided Welch two-sample *t*-test). Thus, for CI-AMPAR-mediated eEPSCs NASPM gave results similar to those obtained with spermine.

In CGCs from GluA2 KO mice, both spermine and NASPM conferred inward rectification (*Figure 7b and c*). However, at +60 mV, while prominent outward currents were seen with spermine in the pipette, with NASPM virtually all current was abolished. Consequently, the $RI_{+60/−60}$ values with spermine (0.29 ± 0.02, n=6) and NASPM (0.04 ± 0.02, n=4) were different (unpaired mean difference −0.25 [−0.31, –0.21]; *p*<0.0001, two-sided Welch two-sample *t*-test). Thus, CP-AMPARs at synapses formed in vivo behave like all other GluA2-lacking CP-AMPARs examined, allowing outward current at positive potentials with 100 µM intracellular spermine but exhibiting near-complete inward rectification with 100 µM intracellular NASPM.

## Discussion

Our results demonstrate that when included in the intracellular recording solution the exogenous polyamine toxins PhTx-433 and PhTx-74 and the toxin analog NASPM cause a selective, voltage-dependent blocks of GluA2-lacking CP-AMPARs. As seen with spermine (*Soto et al., 2007*), the potency of the blockers is reduced when the receptors are associated with the TARP γ2. However, unlike spermine, which is currently widely used as means of identifying CP-AMPARs, NASPM can produce a complete block of outward currents through CP-AMPARs. Furthermore, the block by NASPM remains essentially complete when CP-AMPARs are co-assembled with multiple classes of auxiliary subunits, as demonstrated in expression systems and for both extrasynaptic and synaptic native receptors. These characteristics make the use of intracellular NASPM a valuable new approach to disclose the presence, and relative proportion, of functional CP-AMPARs. Indeed, given that spermine is often added to the pipette solutions when recording EPSCs and other AMPAR-mediated

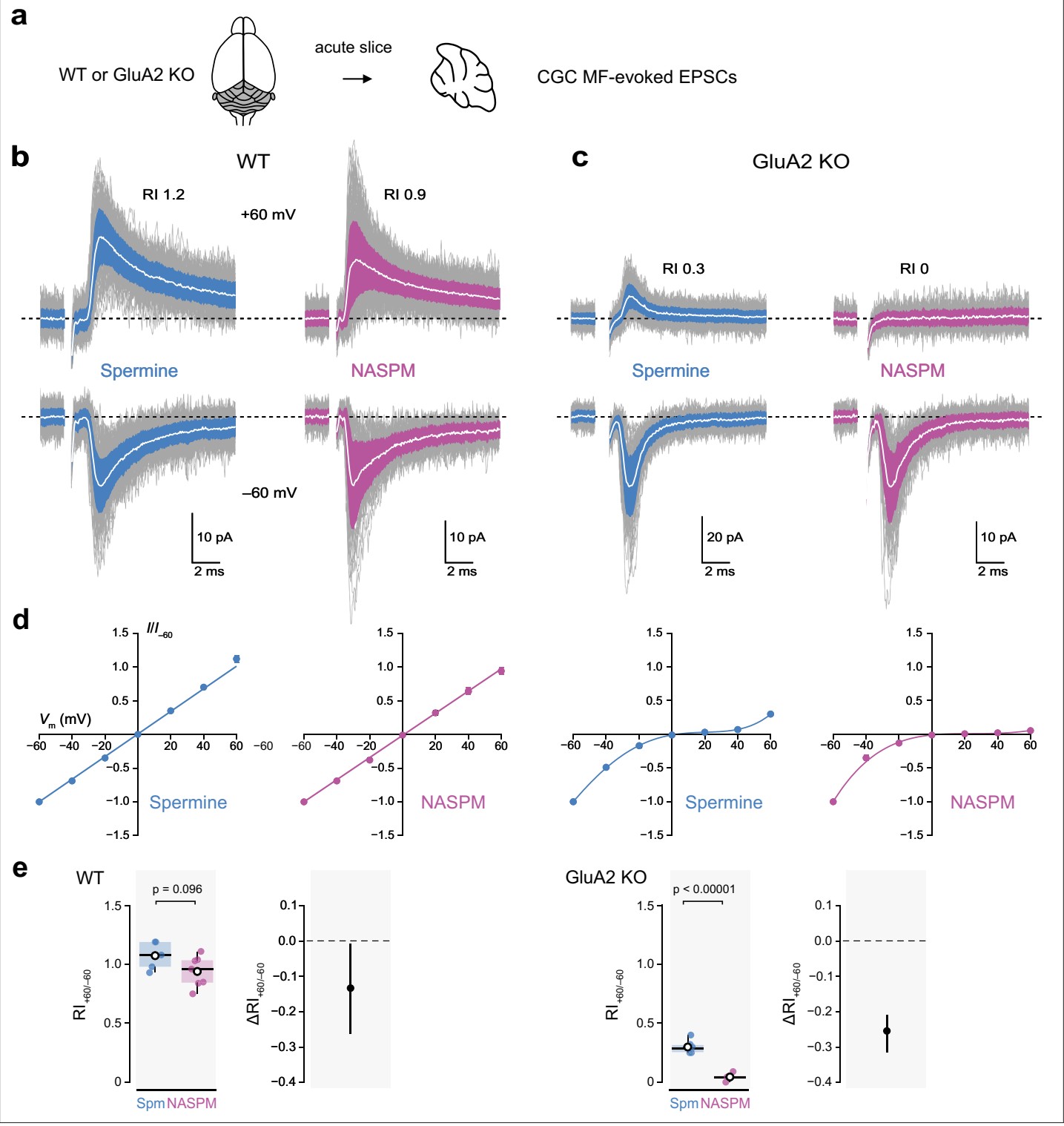

**Figure 7.** Intracellular 1-naphthylacetyl spermine (NASPM) (100 μM) produces full rectification of calcium-permeable AMPA-type glutamate receptor (CP-AMPAR)-mediated MF-eEPSCs. (**a**) Diagrammatic representation of the preparation. (**b**) Representative whole-cell recordings of mossy fiber-evoked EPSCs from wild-type (WT) cerebellar granule cells (CGCs) held at −60 and +60 mV with 100 μM intracellular spermine (blue) or 100 μM intracellular NASPM (purple). Individual mEPSCs are shown in gray, the averaged mEPSCs are shown in white, superimposed on filled areas indicating the standard deviation of each recording. The RI values are of the illustrated records (peak averaged current at +60/peak averaged current at −60 mV). (**c**) Same as a, but for two cells in slices from GluA2-knockout (GluA2 KO) mice. (**d**) Normalized current-voltage (*I-V*) plots for eEPSCs recorded from cells in slices from WT and GluA2 KO mice with 100 μM intracellular spermine (blue) or 100 μM intracellular NASPM (purple). Symbols denote mean values and error bars

*Figure 7 continued on next page*

*Figure 7 continued*

denote the standard error of the mean (WT; n=5 spermine and 7 NASPM. GluA2 KO; n=6 spermine and 4 NASPM). Fits are straight lines (WT) or fifth-order polynomials (GluA2 KO). (**e**) Box-and-whisker and difference plots as described in *Figure 4c and d* for rectification values (RI$_{+60/−60}$) obtained with spermine (Spm) and NASPM from WT and GluA2 KO cells. Indicated p-values are from two-sided Welch two-sample *t*-tests.

The online version of this article includes the following source data for figure 7:

**Source data 1.** MF-eEPSC peak amplitudes, 20–80% rise times and $\tau_{\text{w,decay}}$ values from wild-type (WT) and GluA2-knockout (GluA2 KO) CGCs recorded with intracellular spermine (100 μM) or NASPM (100 μM).

**Source data 2.** Normalized current-voltage (*I-V*) data for peak MF-eEPSCs from wild-type (WT) and GluA2-knockout (GluA2 KO) CGCs recorded with intracellular spermine (100 μM) or NASPM (100 μM).

**Source data 3.** RI$_{+60/−60}$ values for peak MF-eEPSCs from wild-type (WT) and GluA2-knockout (GluA2 KO) CGCs were recorded with intracellular spermine (100 μM) or NASPM (100 μM).

currents solely to allow rectification to be used as a proxy for the presence of CP-AMPARs, a strong case can be made that use of intracellular NASPM should become the new standard approach in the field.

## The mechanism of polyamine toxin block

Cryo-EM structures of activated GluA2/γ2 AMPARs blocked with NASPM show the hydrophobic naphthalene moiety sitting within the electroneutral central vestibule and the polyamine tail extending into the electronegative selectivity filter (*Twomey et al., 2018*). This is in accord with predictions from molecular modeling of the block by various philanthotoxins and adamantane derivatives (*Andersen et al., 2006*; *Tikhonov, 2007*) and suggests that most of the observed receptors were blocked from the 'outside.' Our data show, unsurprisingly, that the voltage-dependence of a block by intracellular NASPM is the inverse of that seen with extracellular NASPM (*Koike et al., 1997*; *Twomey et al., 2018*), which suggests that intracellularly applied NASPM occupies the pore in an 'inverted' orientation, with its polyamine tail in the selectivity filter, but with its bulky head group protruding to the intracellular side of the channel preventing permeation.

We have shown previously that γ2 can reduce spermine potency at CP-AMPARs by around 20-fold (*Soto et al., 2007*), possibly by increasing spermine permeation (*Brown et al., 2018*). In the present experiments, we found that co-expression of γ2 also reduced the potency of non-permeating blockers (7–18-fold). This is consistent with a TARP-dependent reduction in polyamine affinity, perhaps resulting from a reshaping of the selectivity filter (*Brown et al., 2018*; *Soto et al., 2014*), mediated by interactions of TARP TM4 with the AMPAR M2 pore helix (*Herguedas et al., 2022*). One possible caveat to our experiments on recombinant AMPARs is that we established the efficacy of NASPM against CP-AMPARs using exclusively homomeric GluA1$_{\text{flip}}$-containing combinations. However, given that NASPM was also uniformly effective in GluA2 KO CGCs, which are thought to express both flip and flop GluA4 splice forms (*Mosbacher et al., 1994*) and varying degrees of R/G editing (*Lomeli et al., 1994*), it seems unlikely that these posttranscriptional modifications influence the action of intracellular NASPM.

## Advantages of intracellular NASPM for assessing CP-AMPAR contribution

Although intracellular spermine is widely used in voltage-clamp experiments to assess the contribution of CP-AMPARs to the recorded currents, the inclusion in the intracellular solution of 100 μM NASPM offers two key advantages. First, it can produce a selective, use-independent, and complete block of CP-AMPAR-mediated outward current. Second, this block is unaffected by the presence of auxiliary proteins. Together, these characteristics eliminate the difficulties of interpretation that can arise when using spermine.

Furthermore, the full block of CP-AMPARs at positive potentials will allow any limited contribution of CI-AMPARs to be more effectively assessed. With intracellular spermine, an intermediate level of rectification is difficult to interpret, as it could reflect the presence of a mixture of CI- and CP-AMPARs or the presence of CP-AMPARs with various auxiliary proteins. (*Brown et al., 2018*; *Cho et al., 2007*; *Coombs et al., 2012*; *Soto et al., 2007*; *Soto et al., 2009*). With NASPM, however, incomplete rectification can arise only if there is a genuine contribution of CI-AMPARs. Of note, as the spermine block

of CP-KARs is also affected by the auxiliary proteins Neto1 and –2 (*Brown et al., 2016*; *Fisher and Mott, 2012*), NASPM may also find utility in the study of these receptors.

Whilst a principal advantage of NASPM is its insensitivity to the presence of auxiliary subunits, this means that it cannot discriminate between different CP-AMPAR subtypes with potentially different auxiliary subunit content. By contrast, we have previously used the precise degree of rectification seen with spermine to infer the association of auxiliary subunits with native CP-AMPARs (*Soto et al., 2007*; *Soto et al., 2009*; *Studniarczyk et al., 2013*). Thus, we suggest that intracellular NASPM will prove most informative when used in parallel with intracellular spermine – full rectification with NASPM will identify a pure CP-AMPAR population, allowing the degree of rectification with spermine to provide information regarding the potential presence or absence of specific auxiliary subunits. Without the use of NASPM, information from other measures, such as the channel conductance estimated using nonstationary fluctuation analysis (*Bats et al., 2012*; *Studniarczyk et al., 2013*), is needed to make such inferences.

The extracellular application of cationic blockers, such as HPP-SP, NASPM, PhTx-433, or IEM-1460, has been widely used to reveal the presence and roles of CP-AMPARs in neurons, with the advantage that recordings can be made at a fixed negative voltage and the block can be reversible. However, those extracellular blockers that have been examined on CP-AMPAR-mediated synaptic responses in tissue from GluA2 KO mice – HPP-SP, PhTx-433, and IEM-1460 – produce an incomplete (~60–80%) block, even when used at relatively high concentrations (10 or 100 µM) (*Adesnik and Nicoll, 2007*; *Gray et al., 2007*; *Jackson and Nicoll, 2011a*, *Mainen et al., 1998*; *Sara et al., 2011*). In this respect, it is revealing that a low concentration (100 nM) of PhTx-433, which is capable of producing a near complete steady-state block of recombinant GluA2-lacking receptors, has only a minimal effect on GluA2-lacking synaptic responses (*Jackson and Nicoll, 2011a*). This difference in efficacy reflects the different nature of glutamate exposure and the use-dependent nature of the block. This also means that, when used to examine the properties of synaptic responses, the effects can be frequency-dependent and exhibit a slow onset (*Lujan et al., 2019*; *Mainen et al., 1998*; *Zaitsev et al., 2011*). Finally, of course, extracellularly applied drugs can block CP-AMPARs throughout the tissue, not only those in the recorded cell. This can potentially affect glutamate release, either directly or indirectly.

To conclude, our recordings of macroscopic currents from excised patches containing heterologously expressed or native AMPARs, of mEPSCs from dissociated neurons, and of eEPSCs from neurons in brain slices have demonstrated the value of intracellular NASPM as a tool for the functional study of CP-AMPARs. We suggest that intracellular NASPM offers a clear methodological advantage over spermine for experiments on recombinant or native AMPARs where rectification is used to assess the contribution of CP-AMPAR to the current. It enables a straightforward and cell-specific readout of CP-AMPAR presence and the ability to unambiguously assess changes in their prevalence.

## Materials and methods

**Key resources table**

| Reagent type (species) or resource | Designation | Source or reference | Identifiers | Additional information |
|---|---|---|---|---|
| Strain, strain background (*Mus musculus*) | *Gria2^{tm1Rod}*/J, 129 /CD1 | other | | Gift from Ingo Greger, MRC LMB, Cambridge, UK |
| Genetic reagent (*Mus musculus*) | oIMR6780 (*Gria2*) | JAX | JAX:oIMR6780 | wild-type Forward primer GGT TGG TCA CTC ACC TGC TT |
| Genetic reagent (*Mus musculus*) | oIMR6781 (*Gria2*) | JAX | JAX:oIMR6781 | Common primer TCG CCC ATT TTC CCA TAT AC |
| Genetic reagent (*Mus musculus*) | oIMR8444 (*Gria2*) | JAX | JAX:oIMR8444 | Mutant primer GCC TGA AGA ACG AGA TCA GC |
| Cell line (*Homo sapiens*) | HEK293 | ATCC | ATTC:CRL-1573 | |
| Transfected construct (*Rattus norvegicus*) | pIRES-eGFP-GluA1 | doi:10.1126/science.2166337; doi:10.1038/nn.2266 | | *Gria1*; Flip form. Gift from Peter Seeburg (subcloned into pIRES-eGFP) |

*Continued on next page*

*Continued*

| Reagent type (species) or resource | Designation | Source or reference | Identifiers | Additional information |
|---|---|---|---|---|
| Transfected construct (*Rattus norvegicus*) | pIRES-eGFP-GluA2 | doi:10.1126/science.2166337; doi:10.1523/JNEUROSCI.17-01-00058.1997 | | *Gria2*; Q/R and R/G edited flip form. Gift from Peter Seeburg (subcloned into pIRES-eGFP) |
| Transfected construct (*Rattus norvegicus*) | pIRES-eGFP-γ2 | doi:10.1038/1228; doi:10.1038/nn.2266 | | *Cacng2*; Gift from Roger Nicoll, UCSF, USA |
| Transfected construct (*Homo sapiens*) | pIRES-eGFP-γ7 | doi:10.1038/nn.2266 | | *CACNG7*; OriGene Technologies pCMV6-XL4-γ7 (subcloned into pIRES-eGFP) |
| Transfected construct (*Rattus norvegicus*) | pIRES-eGFP-γ8 | doi:10.1083/jcb.200212116; doi:10.1038/nn.2266 | | *Cacng8*; gift from Roger Nicoll, UCSF, USA |
| Transfected construct (*Rattus norvegicus*) | pIRES-eGFP-CNIH2 | doi:10.1126/science.1167852; doi:10.1523/JNEUROSCI.0345–12.2012 | | *Cnih2*; gift from Bernd Fakler, University of Freiburg, Germany (subcloned into pIRES-eGFP) |
| Transfected construct (*Rattus norvegicus*) | pIRES-eGFP-CNIH3 | doi:10.1016/j.neuron.2012.03.034; doi:10.1523/JNEUROSCI.0345–12.2012 | | *Cnih3*; Gift from Bernd Fakler, University of Freiburg, Germany (subcloned into pIRES-eGFP) |
| Transfected construct (*Mus musculus*) | pRK5-CKAMP44a | doi:10.1126/science.1184245 | | *Ckamp44*; Gift from Jakob von Engelhardt, Johannes Gutenberg University, Mainz, Germany |
| Transfected construct (*Mus musculus*) | pRK5-CKAMP59-short | doi:10.7554/eLife.09693.001 | | *Ckamp59*; Gift from Jakob von Engelhardt, Johannes Gutenberg University, Mainz, Germany |
| Transfected construct (*Rattus norvegicus*) | pcDNA3.1-GSG1L | doi:10.1016/j.neuron.2012.03.034; doi:10.1523/JNEUROSCI.2152–15.2015 | | *Gsg1l*; Gift from Bernd Fakler, University of Freiburg, Germany |
| Transfected construct (*Rattus norvegicus*) | pcDNA3-Homer1c-tdDsRed | doi:10.1016/j.neuron.2007.01.030 | | Gift from Daniel Choquet, University of Bordeaux, France |
| Chemical compound, drug | PhTx-433; Spermine tetrahydrochloride | Sigma-Aldrich, Merck Life Science UK Limited, Gillingham, UK | Sigma-Adrich:P207; Sigma-Adrich:85610 | |
| Chemical compound, drug | PhTx-74 | Tocris Bioscience, Bio-Techne Ltd, Abingdon, UK | Tocris:2770 | |
| Chemical compound, drug | NASPM | HelloBio, Bristol, UK | HelloBio:HB0441 | |
| Software, algorithm | IGOR Pro | Wavemetrics, Lake Oswego, Oregon, USA | RRID:SCR_000325 | version 6.35 |
| Software, algorithm | pClamp | Molecular Devices | RRID:SCR_011323 | version 10 |
| Software, algorithm | WinWCP | Strathclyde Electrophysiology Software | RRID:SCR_014713 | version 5.2.7 |
| Software, algorithm | NeuroMatic | http://www.neuromatic.thinkrandom.com | RRID:SCR_004186 | version 2.8 |
| Software, algorithm | R | R Foundation for Statistical Computing | RRID:SCR_001905 | version 4.1.0 |
| Software, algorithm | RStudio Desktop | Posit Software | RRID:SCR_000432 | version 2022.12.0 |

## Materials

Common laboratory chemicals were obtained from Sigma-Aldrich (Merck Life Science UK Limited, Gillingham, UK), as were penicillin, streptomycin, transferrin, insulin, spermine tetrahydrochloride, strychnine, and PhTx-433. Tetrodotoxin (TTX) and PhTx-74 were from Tocris Bioscience (Bio-Techne Ltd, Abingdon, UK). NASPM and bicuculline methiodide were from HelloBio (Bristol, UK). Cyclothiazide, D-AP5, SR-95531 [2-(3-carboxypropyl)–3-amino-6-(4-methoxyphenyl)pyridazinium bromide] and QX-314 [N-(2,6-dimethylphenylcarbamoylmethyl)triethylammonium bromide] were from Tocris

Bioscience or HelloBio. Lipofectamine 2000 and gentamicin were from Invitrogen (Thermo Fisher Scientific, Waltham, MA USA). Basal Medium Eagle (BME), Eagles Minimum Essential Medium (MEM), Dulbecco's Modified Eagle Medium (DMEM), and fetal bovine serum (FBS) were from Gibco (Thermo Fisher Scientific, Waltham, MA USA).

## Mice

Gria2-deficient (GluA2 KO) mice were bred from heterozygous parents ($Gria2^{tm1Rod}$/J / $Gria2^+$) (Jia et al., 1996) on a 129 /CD1 background. Mice were group housed in standard cages and maintained under controlled conditions (temperature 20 ± 2°C; 12 hr light-dark cycle). Food and water were provided ad libitum. Both male and female mice were used for generating primary neuronal cultures or slices. GluA2 KO ($Gria2^{-/-}$) mice were compared with wild-type ($Gria2^{+/+}$) littermates. The genotypes of the pups were determined using PCR analysis using the following Gria2 primers: GGTT GGTCACTCACCTGCTT (wild-type, oIMR6780); TCGCCCATTTTCCCATATAC (common, oIMR6781) and GCCTGAAGAACGAGATCAGC (mutant, oIMR8444). All procedures for the care and treatment of mice were in accordance with the Animals (Scientific Procedures) Act 1986.

## Heterologous expression

Plasmids containing the flip splice variant of rat GluA1 (pIRES-eGFP-GluA1), the Q/R and R/G edited flip splice variant of rat GluA2 (pIRES-eGFP-GluA2), rat γ2 (pIRES-eGFP-γ2), rat CNIH2 (pIRES-eGFP-CNIH2), rat CNIH3 (pIRES-eGFP-CNIH3), rat GSG1L (pcDNA3.1-GSG1L), human γ7 (pIRES-eGFP-γ7), mouse CKAMP44 (pRK5-CKAMP44a), and mouse CKAMP59 (pRK5-CKAMP59-short) were expressed in human embryonic kidney (HEK) 293 cells from ATCC (Manassas, VA, USA; CRL-1573). The authentication of this cell line was provided by ATCC, and the cell line was tested negative for mycoplasma contamination. Cells were grown in DMEM supplemented with 10% FBS, 100 U/ml penicillin, 0.1 mg/ml streptomycin at 37 °C, 5% $CO_2$, and maintained according to standard protocols. Transient transfection was performed using Lipofectamine 2000 according to the manufacturer's instructions. Heteromeric GluA1/2 receptors were expressed using a cDNA ratio of 1:2 GluA1:GluA2. AMPAR/auxiliary subunit combinations had a cDNA ratio of 1:2 GluA1:auxiliary subunit. Cells were split 12–24 hr after transfection and plated on poly-L-lysine coated glass coverslips. Electrophysiological recordings were performed 18–48 hr later.

## HEK293 cell outside-out patch recordings

Patch-clamp electrodes were pulled from borosilicate glass (1.5 mm o.d., 0.86 mm i.d.; Harvard Apparatus, Cambridge, UK) and fire polished to a final resistance of 6–10 MΩ. Voltage ramp experiments were performed using outside-out patches. The 'external' solution contained: 145 mM NaCl, 2.5 mM KCl, 1 mM CaCl₂, 1 mM MgCl₂, and 10 mM HEPES (pH 7.3). For steady-state ramp responses, this solution was supplemented with 300 µM glutamate and 50 µM cyclothiazide. The 'internal' solution contained 125 mM CsCl, 2.5 mM NaCl, 1 mM CsEGTA, 10 mM HEPES, and 20 mM Na₂ATP (pH 7.3 with CsOH) and was supplemented with the indicated concentrations of spermine tetrahydrochloride, NASPM, PhTx-433, or PhTx-74. For fast-application experiments, the 'external' solution contained 145 mM NaCl, 2.5 mM KCl, 2 mM CaCl₂, 1 mM MgCl₂, and 10 mM HEPES (pH 7.3) while the 'internal' solution contained (in mM): 140 mM CsCl, 10 mM HEPES, 5 mM EGTA, 2 mM MgATP, 0.5 mM CaCl₂, and 4 mM NaCl, supplemented with 10 or 100 µM spermine or NASPM as indicated. Recordings were made at 22–25°C using an Axopatch 200 A amplifier (Molecular Devices, USA). Voltage ramp currents were low-pass filtered at 500 Hz and digitized at 2 kHz using an NI USB-6341 (National Instruments) interface with WinWCP (v5.2.7; Strathclyde Electrophysiology Software; http://spider.science.strath.ac.uk/sipbs/software_ses.htm). Recordings of responses to fast glutamate applications were low pass filtered at 10 kHz and digitized at 20 kHz.

Voltage ramps (either −80 mV to +60 mV, −100 mV to +100 mV, or −100 mV to +130 mV) were delivered at 100 mV/s. For each experiment, ramps in control or glutamate plus cyclothiazide solutions were interleaved to allow for leak subtraction. Rapid agonist application was achieved by switching between continuously flowing solutions, as described previously (Soto et al., 2014). Solution exchange was achieved by moving an application tool – made from custom triple-barrelled glass (VitroCom, Mountain Lakes, NJ, USA) – mounted on a piezoelectric translator (PI (Physik Instrumente) Ltd, Bedford, UK). At the end of each experiment, the adequacy of the solution exchange was tested

by destroying the patch and measuring the liquid-junction current at the open pipette (10–90% rise time typically 150–300 µs).

## Neuronal primary culture

Primary cultures were prepared from the cerebella of P7 mice as previously described (*McGee et al., 2015*). After dissociation, the cell suspension was transfected with a pcDNA3-Homer1c-tdDsRed plasmid (*Bats et al., 2007*) by electroporation using the Amaxa Nucleofector 2b device and Amaxa mouse neuron nucleofector kit (Lonza). Neurons were then plated on poly-L-lysine-coated glass coverslips and grown at 37 °C in a humidified 5% $CO_2$-containing atmosphere in BME supplemented with KCl (25 mM final concentration), 20 µg/ml gentamicin, 2 mM L-glutamine, and 10% v/v FBS. After 3 days in vitro, the growing medium was replaced with MEM supplemented with 5 mg/ml glucose, 2 mM glutamine, 20 µg/ml gentamicin, 0.1 mg/ml transferrin, and 0.025 mg/ml insulin. Recordings were performed after 8–11 DIV.

## mEPSC recordings

Cerebellar cells grown on coverslips were visualized using an inverted microscope (IX71; Olympus) equipped with a 40x/0.9 NA objective (Olympus) and excitation and emission filters (Chroma Technology HQ540/40 x and HQ600/50 m) to enable DsRed visualization. We distinguished stellate cells from granule cells by their larger soma and the presence of punctate somatic and dendritic DsRed labeling that differed from the sparse, more distal dendritic, labeling of granule cells.

The extracellular solution, adjusted to pH 7.3 with NaOH, contained: 145 mM NaCl, 2.5 mM KCl, 2 mM $CaCl_2$, 1 mM $MgCl_2$, 10 mM glucose, and 10 mM HEPES. To this, we added 0.5 µM TTX, 20 µM D-AP5, and 20 µM SR-95531 to block voltage-gated sodium channels, NMDA, and $GABA_A$ receptors, respectively. The internal solution, adjusted to pH 7.4 with CsOH, contained: 140 mM CsCl, 10 mM HEPES, 5 mM EGTA, 2 mM MgATP, 0.5 mM $CaCl_2$, 4 mM NaCl, and either 100 µM spermine or 100 µM NASPM. Recordings were performed using an Axopatch 700B amplifier, low pass filtered at 6 kHz, and digitized at 25 kHz using a USB-6341 interface and Igor Pro 6 (v6.35, Wavemetrics, Lake Oswego, Oregon, USA) with NeuroMatic v2.8 and NClamp (http://www.neuromatic.thinkrandom. com) (*Rothman and Silver, 2018*). Patch-clamp electrodes were pulled from borosilicate glass (as for outside-out patch recordings) and had a resistance of 5–7 MΩ. $R_{series}$ was not compensated but was monitored throughout each recording; if a cell showed a>30% change in $R_{series}$ it was excluded from the analysis. For each included cell $R_{series}$ was <18 MΩ and $R_{input}$ was >45 × $R_{series}$. Cells were exposed for 2 min to 200 µM $LaCl_3$ prior to data acquisition to increase mEPSC frequency (*Chung et al., 2008*). Membrane potential was stepped in 10 mV increments from −80 to +60 mV and then from +60 to −80 mV. For the calculation of mEPSC $RI_{+60/−60}$, where both ascending and descending runs were completed, the data were pooled.

## Hippocampal and cerebellar slice preparation

### Hippocampal slices

Mice (P14-P21) were anesthetized with isoflurane and decapitated. The brain was immersed in an ice-cold slicing solution containing 85 mM NaCl, 2.5 mM KCl, 1 mM 1 $CaCl_2$, 1.25 $NaH_2PO_4$, 26 $NaHCO_3$, 75 sucrose, 4 $MgCl_2$, 25 glucose, supplemented with 20 µM D-AP5 to prevent NMDAR-mediated cell damage (pH 7.4 when bubbled with 95% $O_2$/5% $CO_2$). Horizontal slices (250 µm) were cut using a vibratome (Leica VT1200S) and transferred to a submerged holding chamber at 37 °C. The solution was slowly exchanged over 1 hr to an external 'recording' solution containing 125 mM NaCl, 2.5 mM KCl, 1.25 mM $NaH_2PO_4$, 26 mM $NaHCO_3$, 2 mM $CaCl_2$, 1 mM $MgCl_2$, 25 mM glucose, and 20 µM D-AP5 (pH 7.4 when bubbled with 95% $O_2$/5% $CO_2$). Thereafter, the slices were maintained at room temperature prior to recording.

### Cerebellar slices

Parasagittal slices (280 µm) were prepared as for hippocampal slices, except the slicing solution containing 0.5 mM $CaCl_2$, 64 mM sucrose, and 40 µM D-AP5, and a different vibratome was used (Campden 7000smz). Slices were stored in the same solution in a submerged chamber at 32 °C for 30 min and then transferred into recording 'external' solution (as for hippocampal slices, but without D-AP5) for maintenance at room temperature prior to recording.

## DGGC outside-out patch recordings

Hippocampal slices were viewed with an upright microscope (BX51WI; Olympus) using a 60x/0.9 NA objective (Olympus) and oblique illumination. Outside-out patches were pulled from visually identified dentate gyrus granule cells using thick-walled borosilicate glass electrodes. The electrodes had a resistance of 8–10 M$\Omega$ when filled with an internal solution containing: 128 mM CsCl, 10 mM HEPES, 10 mM EGTA, 2 mM $Mg_2ATP$, 0.5 mM $CaCl_2$, 2 mM NaCl, 5 mM TEA, 1 mM QX-314, and either 0.1 mM spermine tetrahydrochloride or 0.1 mM NASPM (pH 7.3 with CsOH; osmolarity 300 ± 20 mOsm/L). Slices were perfused with the external solution containing 20 µM D-AP5 (as described above, under *Hippocampal slices*) to which was added 20 µM bicuculline methobromide and 1 µM strychnine. Currents were recorded at room temperature using an Axopatch 200B amplifier, filtered at 10 kHz, and digitized at 20 kHz using WinWCP software.

Rapid glutamate application was achieved by jumping between a control solution stream and one containing 10 mM glutamate for 100 ms using an application tool made of theta glass (Hilgenberg) mounted on a piezoelectric translator (PI (Physik Instrumente) Ltd, Bedford, UK). Open-tip responses between the control and a diluted solution had 10–90% rise times that were typically 250–350 µs. For calculation or the rectification index ($RI_{+60/-60}$), multiple jumps into glutamate were performed at −60 mV then at +60 mV, before being repeated and currents averaged. For the generation of *I-V* plots, multiple jumps into glutamate were performed at intervals of 10 mV between −60 mV and +60 mV in a randomized sequence. To reduce the impact of noise on the measurement of peak currents, recordings were low-pass filtered offline at 2 kHz.

## Mossy fiber-evoked CGC EPSC recordings

Cerebellar slices were viewed with an upright microscope (BX50WI; Olympus) using a 40x/0.9 NA objective (Olympus) and oblique illumination. Whole-cell recordings were made from visually identified cells in the internal granule cell layer. To record eEPSCs, mossy fibers were stimulated (1 Hz) using constant voltage pulses (10–100 µs; 30–95 V) delivered through a glass electrode filled with external solution and placed in the granule cell layer, typically ~50–100 µm from the soma of recorded granule cell.

The extracellular solution contained: 125 mM NaCl, 2.5 mM KCl, 2 mM $CaCl_2$, 1 mM $MgCl_2$, 25 mM $NaHCO_3$, 1.25 mM $NaH_2PO_4$, 25 mM glucose; pH 7.3 when bubbled with 95% $O_2$/5% $CO_2$. To block NMDA-, $GABA_A$-, and glycine receptors, 20 µM D-AP5, 20 µM SR-95531, and 1 µM strychnine were added to the external solution. The internal solution, adjusted to pH 7.3 with CsOH, contained: 128 mM CsCl, 10 mM HEPES, 10 mM EGTA, 2 mM $Mg_2ATP$, 0.5 mM $CaCl_2$, 2 mM NaCl, 5 mM TEA, 1 mM QX-314, and either 0.1 mM spermine tetrahydrochloride or 0.1 mM NASPM (osmolarity 300 ± 20 mOsm/L). Recordings were acquired with an Axopatch 200B amplifier, low pass filtered at 5 kHz, and digitized at 20 kHz using a Digidata 1440 A interface with pClamp 10 software (Molecular Devices). Electrodes were pulled from borosilicate glass (as for outside-out patch recordings) and had a resistance of 5–7 M$\Omega$. $R_{series}$ was not compensated but was monitored throughout each recording; if a cell showed a >30% change in $R_{series}$ it was excluded from the analysis. For each included cell $R_{series}$ was <35 M$\Omega$ and $R_{input}$ was >142 × $R_{series}$. Cells were exposed for 2 min to 200 µM $LaCl_3$ prior to data acquisition to increase mEPSC frequency (*Chung et al., 2008*). Membrane potential was stepped in 10 mV increments from −80 to +60 mV and then from +60 to −80 mV. For the calculation of mEPSC $RI_{+60/-60}$, where both ascending and descending runs were completed, the data were pooled.

## Data analysis

### Analysis of HEK cell recordings

Records were analyzed using IGOR Pro with NeuroMatic. To quantify the rectification seen with ramp *I-V* relationships, the rectification index ($RI_{+60/-60}$) was calculated as the ratio of the current at +60 mV and −60 mV (average of 15 data points spanning each voltage). To generate conductance-voltage (*G-V*) curves the reversal potential for each leak-subtracted *I-V* curve was calculated to ascertain the driving force. The resultant data were normalized, averaged, then converted to conductances. To account for the polyamine-independent outward rectification of AMPARs, the conductance values were divided by those obtained in the polyamine-free condition. For currents that displayed inward rectification only, *G-V* curves were fitted with the Boltzmann equation:

$$G = G_{max} \left( \frac{1}{1+exp\left(\frac{V_m - V_b}{k_b}\right)} \right),$$

where $G_{max}$ is the conductance at a sufficiently hyperpolarised potential to produce full relief of polyamine block, $V_m$ is the membrane potential, $V_b$ is the potential at which 50% of block occurs, and $k_b$ is a slope factor describing the voltage dependence of block (the membrane potential shift necessary to cause an e-fold change in conductance). For currents that displayed double rectification, $G$-$V$ curves were fitted with a double Boltzmann equation which contains equivalent terms for voltage-dependent permeation (p) (**Panchenko et al., 1999**):

$$G = G_{max} \left( \frac{1}{1+exp\left(\frac{V_m - V_b}{k_b}\right)} \right) + G_{max,p} \left( \frac{1}{1+exp\left(\frac{V_m - V_p}{k_p}\right)} \right)$$

The slopes of the fits were not constrained. Data from GluA1 with 1 µM NASPM were excluded due to excessive scatter and poor fit. Steeper relationships at more positive potentials in the presence of NASPM (**Figure 2a** GluA1/γ2) likely reflect the modest accumulation of block during the voltage ramp. GluA1 Plots of $V_b$ against the log of the polyamine concentration were fitted with a linear function, the x-axis intercepts giving the voltage-independent affinity ($IC_{50, 0 mV}$).

Current decays following fast applications of glutamate (10 mM, 100 ms) at positive and negative potentials were described by single or double exponential fits. In the latter case, the weighted time constant of decay ($\tau_{w,decay}$) was calculated according to:

$$\tau_{w,decay} = \tau_f \left( \frac{A_f}{A_f + A_s} \right) + \tau_s \left( \frac{A_s}{A_f + A_s} \right),$$

where $A_f$ and $\tau_f$ are the amplitude and time constant of the fast component and $A_s$ and $\tau_s$ are the amplitude and time constant of the slow component. For each condition, the values of $\tau_{decay}$ (from single exponential fits) and $\tau_{w,decay}$ (from double exponential fits) were pooled.

Double pulse experiments (100 ms glutamate applications at intervals of 150 ms to 7.9 s 4–0.125 Hz) were used to assess the recovery of peak responses following the removal of glutamate. As expected, some residual desensitization was apparent at the shortest of the intervals in all conditions (**Coombs et al., 2017**). Thus, the magnitude of the second pulse reflected recovery from desensitization and, at +60 mV with 10 µM NASPM, the relief of the NASPM block. In the latter case, the recovery of the second peak was fitted with a biexponential function, as above.

## mEPSC analysis

mEPSCs were detected using an amplitude threshold crossing method based on the algorithm of **Kudoh and Taguchi, 2002** (NeuroMatic). The standard deviation of the background noise at +60 mV (range 2.4–6.4 pA; 3.5 ± 0.4 pA, n=12) was determined by fitting a single-sided gaussian to an all-point histogram from a 500ms stretch of record. For each cell, the same detection threshold (2–3 x the standard deviation at +60 mV) was used at both −60 and +60 mV. At each voltage, the mEPSC frequency was determined as the total number of mEPSCs detected/record length, and a mean mEPSC waveform was constructed from those events that displayed a monotonic rise and an uncontaminated decay. One cell recorded with intracellular spermine was excluded from the analysis as <10 events were detected during 40 s recording at −60 mV. For each of the 12 other cells, detected events were aligned on their rising phase before averaging. The rectification index was then calculated as:

$$RI_{+60/-60} = \left( \left| \bar{I}_{+60} \right| \times f_{+60} \right) / \left( \left| \bar{I}_{-60} \right| \times f_{-60} \right)$$

where $\bar{I}$ is the amplitude of the mean mEPSC and $f$ is the frequency of mEPSCs at the indicated voltage. For each averaged mEPSC we determined the 20–80% rise time and fitted the decay with a double exponential to obtain $\tau_{w,decay}$ (as described for HEK cell agonist-evoked currents).

## MF-eEPSC analysis

eEPSCs were evoked at multiple voltages between −80 and +60 mV. Average waveforms were generated from responses to ~100 stimuli and the peak amplitude was determined. The rectification index ($RI_{+60/-60}$) was calculated for each cell as the absolute peak of the averaged current at +60 mV divided

by the absolute peak of the averaged current at −60 mV. The 20–80% risetime and $\tau_{w,decay}$ (from double exponential fits) of averaged currents at −60 and +60 mV were also determined.

## Data presentation and statistics

Summary data are presented in the text as mean ± standard error of the mean from $n$ measures, where $n$ represents the number of biological replicates (number of cells or patches). Estimates of paired or unpaired mean differences and their bias-corrected and accelerated 95% confidence intervals from bootstrap resampling are presented as effect size [lower bound, upper bound]. Error bars on graphs indicate the standard error of the mean or, where indicated, the 95% confidence interval. Box-and-whisker plots indicate the median (black line), the 25–75th percentiles (box), and the 10–90th percentiles (whiskers); filled circles are data from individual patches/cells and open circles indicate means. Specific analyses and statistical tests are described in the text. The source data are given in the Source Data files for each relevant figure. Comparisons were performed using paired or unpaired two-sided Welch two-sample $t$-tests that do not assume equal variance (normality was not tested statistically but gauged from density histograms and/or quantile-quantile plots). Exact p-values are presented to two significant figures, except when p<0.0001. Statistical tests were performed using R (version 4.1.0, the R Foundation for Statistical Computing, http://www.r-project.org/) and RStudio (version 2022.12.0, Posit Software). No statistical test was used to predetermine sample sizes. No randomization was used.

## Acknowledgements

This work was supported by an MRC Programme Grant (MR/T002506/1) to MF and SGCC.

## Additional information

### Funding

| Funder | Grant reference number | Author |
| --- | --- | --- |
| Medical Research Council | MR/T002506/1 | Mark Farrant<br>Stuart G Cull-Candy |

The funders had no role in study design, data collection and interpretation, or the decision to submit the work for publication.

### Author contributions

Ian Coombs, Conceptualization, Formal analysis, Investigation, Visualization, Writing – original draft, Writing – review and editing; Cécile Bats, Craig A Sexton, Dorota Studniarczyk, Formal analysis, Investigation, Writing – review and editing; Stuart G Cull-Candy, Conceptualization, Funding acquisition, Writing – original draft, Writing – review and editing; Mark Farrant, Conceptualization, Formal analysis, Funding acquisition, Visualization, Writing – original draft, Writing – review and editing

### Author ORCIDs

Ian Coombs ⓘ http://orcid.org/0000-0003-1006-7471
Cécile Bats ⓘ http://orcid.org/0000-0002-2246-7248
Craig A Sexton ⓘ http://orcid.org/0000-0002-7617-8361
Dorota Studniarczyk ⓘ http://orcid.org/0000-0003-4109-7956
Stuart G Cull-Candy ⓘ http://orcid.org/0000-0002-0831-8326
Mark Farrant ⓘ http://orcid.org/0000-0002-9918-0376

### Ethics

All procedures for the care and treatment of mice were in accordance with the Animals (Scientific Procedures) Act 1986 (licences PPL 70/8526 and P4114FCF5) and institutional animal care and use committee (IACUC) protocols at University College London.

### Decision letter and Author response

Decision letter https://doi.org/10.7554/eLife.66765.sa1

Author response https://doi.org/10.7554/eLife.66765.sa2

## Additional files

### Supplementary files
• Transparent reporting form

### Data availability
All data generated during this study are included in the manuscript and supporting files. Source data files have been provided for Figures 1, 2, 3, 4, 5, 6 and 7.

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
