## [Editor Report]

AMPA-type glutamate receptors that lack the GluA2 subunit are calcium-permeable and contribute to calcium influx in plasticity and disease. The authors of this Tools and Resources manuscript describe a method for evaluating the presence of GluA2-lacking receptors using intracellular NASPM that avoids complications related to auxiliary subunits that affect biophysical properties. The compelling results provide a valuable new approach for unambiguously differentiating the presence of Ca-permeable and -impermeable AMPA receptors.

---

## [Decision Letter]

**Decision letter after peer review:**

Thank you for submitting your article "Intracellular NASPM allows an unambiguous functional measure of GluA2-lacking calcium-permeable AMPA receptor prevalence" for consideration by *eLife*. Your article has been reviewed by 4 peer reviewers, one of whom is a member of our Board of Reviewing Editors, and the evaluation has been overseen by John Huguenard as the Senior Editor. The following individuals involved in review of your submission have agreed to reveal their identity: Lonnie Wollmuth (Reviewer #2).

Calcium-permeable AMPA receptors (CP-AMPARs) have been shown to have important roles in modulating many aspects of neuronal function. Here the authors demonstrate that complete block of CP-AMPARs, with no apparent effect on CI-AMPARs, can be achieved by intracellular application of the polyamine NASPM. Importantly, the authors provide evidence that this block is not affected by the presence of auxiliary subunits, one of the key caveats regarding prior interpretations of the effects of polyamines and the roles of CP-AMPARs. The reviewers agree that the data supporting the claims of intracellular NASPM block of CP-AMPARs are rigorous and convincing, and overall the presentation of the manuscript is clear. While there is enthusiasm for the question and approach, there is also consensus that the authors have not yet provided sufficient evidence that this new approach allows an unambiguous functional measure of calcium-permeable AMPA receptor prevalence in an intact preparation.

Essential revisions:

1) A demonstration of the usefulness of the approach under more complex conditions would strengthen the strong conclusion about its utility to differentiate between CI and CP AMPARs.

2). There are a number of limitations that should be acknowledged in the discussion, including the lack of data on unexplored auxiliary subunits (see recommendations to authors).

*Reviewer #1 (Recommendations for the authors):*

Additional evidence of the usefulness of the approach under more complex conditions would strengthen the conclusions about its utility. The authors describe some specific uses in the discussion, but a direct demonstration would be more compelling. Differentiation of GluR1 and GluR2-containing receptors is shown only in Figure 1, the remaining figures describe the block at GluR1 in the absence of GluR2-containing AMPARs.

To fully support the applicability of this new approach, the authors could demonstrate the use of it to distinguish the CP and CI-AMPAR content in intact neurons (i.e. WT stellate and perhaps another cell type with different expected ratios of CP and CI AMPAR content).

One wonders what is the "safety factor" of the approach, since spermine will also cause rectification of GluR2-containing AMPARs at high intracellular concentrations. Is this also the case for NASPM? There is discussion that touches on this point of NASPM extracellular block (related to the Twomey reference), but directly addressing this question as in Figure 2 would be useful.

*Reviewer #2 (Recommendations for the authors):*

I have no major criticisms of the work. It was very readable and the figures are laid out extremely nicely. The experiments and analysis are straightforward and fairly standard.

I am convinced that 100 uM NASPM selectivity blocks CP-AMPARs and that it blocks better than spermine - very nice biophysics. For this reason, the authors argue that NASPM will be a better tool to study CP-AMPARs than spermine because of its properties - blocks fully CP-AMPARs independent of associated auxiliary subunits. In fact, while this work is rigorously done, this would probably be the main argument for inclusion in eLife. But is it a subtle improvement or a dramatic one? For the reader to be fully convinced of the utility of NASPM, it might be useful to show experiments where one can really sees the difference between spermine and NASPM. In another neuronal type, perhaps in slices (cerebellum or hippocampus) where auxiliary subunit presence is variable. Something to show that NASPM really is going to be a step above spermines.

*Reviewer #3 (Recommendations for the authors):*

It would be outside the scope of one manuscript to address all possibilities regarding the selectivity of intracellular NASPM for CP-AMPARs. However, these should be briefly discussed as important caveats. These could include: (1) other TARPS not explored (g8, prominent in hippocampus), (2) TARP-TARP interactions (very speculative), (3) interactions with other cation channels, especially KARs, NMDARs, and perhaps voltage-gated ion channels, (4) all permutations of flip/flop R/G editing (wow, highly speculative) and (5) wash-out (once applied, is it possible to remove?).

Additional comments that are largely cosmetic:

1) l 25, consider "…unlike spermine, completely blocks outward synaptic…".

2) l65, consider "…Kumar et al., 2002; Stubblefield and Benke, 2010; …".

3) l68, consider "…Sanderson et al., 2016, 2018…".

4) l87, consider "…could theoretically reflect…" as I do not think there is a reference to support this, though it should be noted as done by the authors.

5) Figure 2. As the typical way of presenting the data in panel a is with an IV plot (and this would be a useful comparison for future work), consider adding a panel next to (or part of) panel a (while without g2 is seen in Figure 1C, the simultaneous comparisons will be striking) with spermine 100um vs NASPM 100uM. I think this will hammer the notion that intracellular spermine is adequate, especially when TARPS are present.

6) Figure 3d. Perhaps esoteric (but related to 3f), is the voltage-dependent alteration of tau-decay use dependent? Was each measurement done at successively more positive potentials? I am wondering how the interplay of closed-blocked versus closed-unblocked/open-blocked kinetics are interacting.

7) L287-289: agree, but this not necessarily shown.

*Reviewer #4 (Recommendations for the authors):*

The reviewer thinks that NASPM alone is challenging to conclude CP- and CI-AMPARs because NASPM sensitivity and calcium permeability could be segregated in a particular combination of AMPAR subunits and auxiliary subunits. Considering NASPM is known as a ca^2+^-permeable AMPA antagonist (from the Tocris website), it is expected to test NASPM sensitivity more systematically or explain why CI-AMPAR is insensitive to NASPM structurally (though a related topic is discussed).

Specifically:

Because the authors tested a limited combination of receptors and auxiliary subunits, it is difficult to conclude whether NASPM blocks all CP-AMPAR unambiguously. At least, structural models should be provided.

Slopes of the conductance-voltage relationships are changed upon TARPg-2 co-expression or different concentrations of NASPM. This should be explained.

This manuscript provides much important information regarding NASPM action on AMPAR biophysics. I wonder if the AMPAR kinetics' model helps to understand how NASPM modulates AMPAR kinetics besides NASPM blocking AMPAR activity.

Figure 1b and c, please confirm these with error bars. If these traces do not have error bars, it would be helpful to see error bars.

Figure 2a, is any explanation to explain changes in slopes?

Figure 3, can NASPM modulation of various AMPAR kinetics be modeled? This helps to understand its MOA.

Figure 3a and 4a, actual currents are tenfold different. Does this suggest that spermine reduces the peak amplitude substantially?

Figure 5. The author needs to evaluate NASPM effects on mEPSC changes in wild-type neurons to examine CP-AMPAR independent effects of NASPM in neurons.

---

## [Author Response]

Essential revisions:1) A demonstration of the usefulness of the approach under more complex conditions would strengthen the strong conclusion about its utility to differentiate between CI and CP AMPARs.

We have performed new experiments using acute hippocampal and cerebellar slices from wild-type and GluA2^−/−^­ mice to compare the actions of spermine and NASPM on two additional neuronal types (with different GluA and auxiliary protein complements). These experiments demonstrate the clear advantage conferred by using NASPM rather than spermine for differentiating CI- and CP-AMPARs. The details are presented in our responses to the individual Reviewers (below).

2). There are a number of limitations that should be acknowledged in the discussion, including the lack of data on unexplored auxiliary subunits (see recommendations to authors).

We have performed additional experiments examining the actions of spermine and NASPM on recombinant AMPARs expressed with a much wider range of auxiliary proteins, including in combinations thought to be present in neurons. In the revised manuscript we have also acknowledged and discussed the remaining limitations highlighted by the Reviewers. Again, the details are presented below in our responses to the individual Reviewers.

Reviewer #1 (Recommendations for the authors):Additional evidence of the usefulness of the approach under more complex conditions would strengthen the conclusions about its utility. The authors describe some specific uses in the discussion, but a direct demonstration would be more compelling. Differentiation of GluR1 and GluR2-containing receptors is shown only in Figure 1, the remaining figures describe the block at GluR1 in the absence of GluR2-containing AMPARs.1.1. To fully support the applicability of this new approach, the authors could demonstrate the use of it to distinguish the CP and CI-AMPAR content in intact neurons (i.e. WT stellate and perhaps another cell type with different expected ratios of CP and CI AMPAR content).

We appreciate the Reviewer’s suggestion to examine more complex conditions/‘intact’ neurons. To this end, we made recordings from hippocampal dentate gyrus granule cells (DGGCs; extrasynaptic receptors in somatic outside-out patches) and cerebellar granule cells (CGCs; mossy-fibre evoked EPSCs) in acute brain slices. In both cases we used slices from wild-type and GluA2^−/–^ mice. For wild-type CGCs, which express purely CI‑AMPARs, eEPSC peak *I-V* relationships were linear in both spermine and NASPM (100 μM) confirming that in neurons, as with recombinant receptors, NASPM at this concentration does not block CI-AMPARs. Conversely, in both CGCs and DGGCs from GluA2^−/–^ mice (containing purely CP-AMPARs), unlike spermine, NASPM produced complete rectification. Thus, NASPM is superior to spermine in definitively assessing CI-/CP-AMPAR content. For wild-type DGGCs, NASPM and spermine gave RI values indicative of a mixed CI- and CP-AMPAR population. In this case, despite the enhanced rectification in NASPM of any CP-AMPAR-mediated component, there was overlap of the observed RIs with spermine and NASPM due to the scatter resulting from the inevitable variation in the CI-/CP-AMPAR prevalence across patches. Therefore the greater the expected proportion of CP‑AMPARs, the greater the benefit of using NASPM.

Of note, in cases where a mixed CP- and CI-AMPAR population is expected (such as in the case of wild-type DGGCs) observing outward current at +60 mV in the presence of NASPM (as we did) allows definitive identification of the presence of CI-AMPARs, whereas interpreting corresponding data obtained with spermine will always involve a measure of uncertainty.

1.2. One wonders what is the "safety factor" of the approach, since spermine will also cause rectification of GluR2-containing AMPARs at high intracellular concentrations. Is this also the case for NASPM? There is discussion that touches on this point of NASPM extracellular block (related to the Twomey reference), but directly addressing this question as in Figure 2 would be useful.

We thank the Reviewer for raising this point. We have made additional recordings comparing the actions of spermine and NASPM on CI-AMPARs (GluA1/2/γ2). As raised by the Reviewer, at the highest concentration tested (400 μM) spermine resulted in a modest inward rectification, as did NASPM. Seeing that 100 μM NASPM can confer complete rectification of CP-AMPARs, we would suggest that concentrations higher than this should be avoided as they may begin to inhibit CI-AMPAR currents. We have added this new data to the revised Results section.

Reviewer #2 (Recommendations for the authors):I have no major criticisms of the work. It was very readable and the figures are laid out extremely nicely. The experiments and analysis are straightforward and fairly standard.I am convinced that 100 uM NASPM selectivity blocks CP-AMPARs and that it blocks better than spermine - very nice biophysics. For this reason, the authors argue that NASPM will be a better tool to study CP-AMPARs than spermine because of its properties - blocks fully CP-AMPARs independent of associated auxiliary subunits. In fact, while this work is rigorously done, this would probably be the main argument for inclusion in eLife. But is it a subtle improvement or a dramatic one? For the reader to be fully convinced of the utility of NASPM, it might be useful to show experiments where one can really sees the difference between spermine and NASPM. In another neuronal type, perhaps in slices (cerebellum or hippocampus) where auxiliary subunit presence is variable. Something to show that NASPM really is going to be a step above spermines.

We thank the Reviewer for their positive comments. We have performed several new experiments to address the points raised. We have now examined AMPAR-mediated currents from acute cerebellar and hippocampal slices from both wild-type and GluA2^−/–^ mice. We find that regardless of the preparation, NASPM is superior to spermine as a tool for identifying CP-AMPARs. We acknowledge that with mixed CP- and CI-AMPARs (wild-type DGGCs) the improvement offered by NASPM can be considered more “subtle” as the enhanced rectification of the CP-component with NASPM becomes lost in the scatter resulting from variation in the CI-/CP-AMPAR balance. However, the greater the expected proportion of CP-AMPARs, the greater the benefit of using NASPM, and in this condition (exemplified by GluA2^−/–^ slices) NASPM provides a “dramatic” improvement. Moreover, as we highlight in the Discussion, in CP-AMPAR-rich conditions, performing parallel recordings with NASPM and spermine can allow an assessment of the auxiliary proteins present through comparison of the extent of the NASPM- and spermine-induced rectification.

Reviewer #3 (Recommendations for the authors):3.1. It would be outside the scope of one manuscript to address all possibilities regarding the selectivity of intracellular NASPM for CP-AMPARs. However, these should be briefly discussed as important caveats. These could include: (1) other TARPS not explored (g8, prominent in hippocampus), (2) TARP-TARP interactions (very speculative), (3) interactions with other cation channels, especially KARs, NMDARs, and perhaps voltage-gated ion channels, (4) all permutations of flip/flop R/G editing (wow, highly speculative) and (5) wash-out (once applied, is it possible to remove?).

We thank the Reviewer for their positive comments.

Regarding points 1 and 2, we performed new experiments and now examine a broader range of AMPAR auxiliaries, including additional TARP, cornichon and CKAMP/shisa proteins. We have also looked at combinations of auxiliaries (CKAMP44/γ8, CNIH2/γ8) – the benefits of NASPM remain apparent regardless of auxiliary or combination of auxiliaries. Moreover, we also addressed these points by examining additional native receptors known to contain different auxiliary subunit combinations.

Regarding point 3, this is certainly an issue worthy of future consideration. Thus, NASPM may well prove to be a useful tool for examining other polyamine-sensitive channels. In this regard, we have added a comment concerning KARs to the Discussion – like CP-AMPARs, CP-KARs have auxiliary proteins that have been shown to reduce spermine-dependent rectification.

Regarding point 4 and the potential effects of different flip/flop splicing and R/G editing – unlike the association with different transmembrane auxiliary proteins, these processes affect regions of the receptor distant from the pore and would not be expected to differentially influence block by spermine or NASPM. Moreover, given that NASPM was uniformly effective in causing full rectification in GluA2^−/−^ cerebellar granule cells, despite the expected presence of both flip and flop isoforms (GuA4_i_ and GluA4_o_; Mosbacher et al. *Science* 1994) and both R/G edited isoforms (Lomeli et al. *Science* 1994) argues that these posttranscriptional modifications do not modify the action of NASPM. We have added a brief comment on this to the Discussion

Regarding point 5, NASPM wash-out – as shown in Figure 3, NASPM can slowly unbind from the closed channel so wash-out should be possible. This may be of relevance (and could be addressed directly) only when recording from inside-out patches or making successive whole-cell recordings from a given cell with different internal solutions.

Additional comments that are largely cosmetic:1) l 25, consider "…unlike spermine, completely blocks outward synaptic…".2) l65, consider "…Kumar et al., 2002; Stubblefield and Benke, 2010; …".3) l68, consider "…Sanderson et al., 2016, 2018…".4) l87, consider "…could theoretically reflect…" as I do not think there is a reference to support this, though it should be noted as done by the authors.

For all points (1-4) we have changed the text and added selected references, as suggested.

5) Figure 2. As the typical way of presenting the data in panel a is with an IV plot (and this would be a useful comparison for future work), consider adding a panel next to (or part of) panel a (while without g2 is seen in Figure 1C, the simultaneous comparisons will be striking) with spermine 100um vs NASPM 100uM. I think this will hammer the notion that intracellular spermine is adequate, especially when TARPS are present.

We thank the Reviewer for this suggestion but have chosen not to include *I-V* relationships here. Figure 2 specifically concerns blocker potencies, and this is best represented using *G-V* relationships. Moreover, equivalent *I-V* relationships in the presence of γ2 (albeit with rapid application rather than ramps) are already shown in Figure 4a. Parenthetically, we presume that when stating that ‘intracellular spermine is adequate’ the Reviewer means ‘inadequate’.

6) Figure 3d. Perhaps esoteric (but related to 3f), is the voltage-dependent alteration of tau-decay use dependent? Was each measurement done at successively more positive potentials? I am wondering how the interplay of closed-blocked versus closed-unblocked/open-blocked kinetics are interacting.

We thank the Reviewer for raising this important point, which certainly should have been included in the original text. Due to the rapidly accumulating block seen at positive potentials, all currents measured using fast jumps at positive potentials were preceded by a jump at negative voltages to ‘flush out’ NASPM from any blocked channels. In so doing we hoped to produce a blank slate for the assessment of peak current inhibition and desensitization kinetics i.e. make each application from the same ‘closed-unblocked’ starting point. We have now added this information to the legend of Figure 3.

7) L287-289: agree, but this not necessarily shown.

We thank the Reviewer for raising this. This text has been clarified and moved from the legend to the main text.

Reviewer #4 (Recommendations for the authors):4.1. The reviewer thinks that NASPM alone is challenging to conclude CP- and CI-AMPARs because NASPM sensitivity and calcium permeability could be segregated in a particular combination of AMPAR subunits and auxiliary subunits. Considering NASPM is known as a ca^2+^-permeable AMPA antagonist (from the Tocris website), it is expected to test NASPM sensitivity more systematically or explain why CI-AMPAR is insensitive to NASPM structurally (though a related topic is discussed).

The Reviewer is correct to state that NASPM is a recognised blocker of CP-AMPARs. However, this is conventionally applied from the outside, blocking at negative voltages with strong use-dependence. Here we have presented the actions of intracellular NASPM. As described in our responses to Reviewer #3, in the revised manuscript we have systematically examined a wider range of auxiliary proteins and have included new data on the actions of NASPM on CI-AMPARs.

Specifically:4.2. Because the authors tested a limited combination of receptors and auxiliary subunits, it is difficult to conclude whether NASPM blocks all CP-AMPAR unambiguously. At least, structural models should be provided.

As described in the response to Reviewer #3, we have carried out additional experiments examining a much broader range of AMPAR auxiliaries and auxiliary combinations – γ2, γ7, γ8, CNIH2, CNIH3, CKAMP59, GSG1L, γ8/CNIH2 and γ8/CKAMP44. These experiments show that the benefits of NASPM persist in all conditions examined. Nevertheless, we acknowledge the somewhat incautious wording of the original title and have revised this to “Enhanced functional detection of synaptic calcium-permeable AMPA receptors using intracellular NASPM”. Regarding structures, we already cite Twomey et al. (2018) in which both a cryo-EM structure of GluA2(Q)/γ2 with NASPM, and a structural model showing spermine in the pore is presented. It is clear from both aspects of that study, that the pore is fully blocked by the polyamine, and that presence of the pore-loop arginine in edited GluA2 (which produces CI-AMPARs) will be incompatible with high affinity NASPM binding. We therefore feel there is little to be gained by replicating these observations and have chosen not to make this addition.

4.3. Slopes of the conductance-voltage relationships are changed upon TARPg-2 co-expression or different concentrations of NASPM. This should be explained.

We thank the Reviewer for raising this issue, prompting us to reexamine the question of *G-V* slopes. For GluA1 with 1 μM NASPM, the fit was indeed noticeably less steep than the fits for other NASPM concentrations. In this case, the fit was likely impacted by the data around 0 mV, a region which naturally contains a large degree of scatter introduced when converting from current to conductance. We included the original fit as the *V*_b_ value obtained (the only variable we subsequently use) was largely unchanged even if the slope was manually constrained to be steeper. Nonetheless, we acknowledge that the fit of this lowest NASPM condition distracted from the main message, and we have now excluded this (and excluded the corresponding *V*­_b_ value from our derivation of *IC*_50 0 mV_). As to the different slopes observed for GluA1/γ2 with NASPM (e.g., 500 μM vs 100 or 10 μM), it is likely that the nature of the ramp protocol allowed a degree of accumulation of block at positive potentials, hence the steepening of the *G-V* for the lower concentrations (comment added to Methods).

4.4. This manuscript provides much important information regarding NASPM action on AMPAR biophysics. I wonder if the AMPAR kinetics' model helps to understand how NASPM modulates AMPAR kinetics besides NASPM blocking AMPAR activity.

The only apparent NASPM modulation of AMPAR kinetics we have shown is the voltage-dependent speeding of current decay during 100 ms applications of glutamate with 10 μM intracellular NASPM (Figure 3c,d). The slowing of decay kinetics in the absence of polyamine which simply reflects desensitization in this case is incidental to our experiment and is likely explained by an increased prevalence of receptors in the high-open probability mode at positive potentials (Prieto et al., *Nat. Neurosci.* 2010). The fact that NASPM has no effect on kinetics at negative potentials, while the speeding at positive potentials mirrors the voltage-dependence of block strongly suggests that there is no effect on kinetics independent of pore block.

4.5. Figure 1b and c, please confirm these with error bars. If these traces do not have error bars, it would be helpful to see error bars.

Traces in Figure 1b are ramp-evoked currents from individual cells. Data in Figure 1c are averaged *I-V* curves across multiple (3–8) cells. As requested, we have now added shading representing the standard error of the mean to these plots and amended the figure legend accordingly.

4.6. Figure 2a, is any explanation to explain changes in slopes?

This is addressed in the response to point 4.3.

4.7. Figure 3, can NASPM modulation of various AMPAR kinetics be modeled? This helps to understand its MOA.

This has been addressed in the response to point 4.4.

4.8. Figure 3a and 4a, actual currents are tenfold different. Does this suggest that spermine reduces the peak amplitude substantially?

This was merely a reflection of the rather high variation in response amplitude always seen with outside-out patch recording (a consequence of variability in electrode tip size, receptor expression and distribution) and not of any population-specific trend. Thus, for GluA1/γ2, with both 10 and 100 μM spermine or NASPM there was a very wide range of 10 mM glutamate-evoked currents amplitudes at −80 mV:

**Author response table 1. sa2table1:** 

NASPM 100 μM:	Mean −153 pA	Range −20 to −401 pA
Spermine 100 μM:	Mean −54 pA	Range −19 to −97 pA
NASPM 10 μM:	Mean −118 pA	Range −12 to −458 pA
Spermine 10 μM:	Mean −105 pA	Range −16 to −690 pA

4.9. Figure 5. The author needs to evaluate NASPM effects on mEPSC changes in wild-type neurons to examine CP-AMPAR independent effects of NASPM in neurons.

This is an important point that we have addressed in new recordings from dentate gyrus granule cells and cerebellar granule cells in slices from wild-type mice (Figure 5 and Figure 7).